# Single-cell transcriptomics reveals immune response of intestinal cell types to viral infection

Sergio Triana[1,2,†] iD, Megan L Stanifer[3,4,†] iD, Camila Metz-Zumaran[5] iD, Mohammed Shahraz[1], Markus Mukenhirn[5], Carmon Kee[4,5] iD, Clara Serger[1], Ronald Koschny[6], Diana Ordoñez-Rueda[7], Malte Paulsen[7], Vladimir Benes[8], Steeve Boulant[4,5,*] iD & Theodore Alexandrov[1,9,10,**] iD

## Abstract

Human intestinal epithelial cells form a primary barrier protecting us from pathogens, yet only limited knowledge is available about individual contribution of each cell type to mounting an immune response against infection. Here, we developed a framework combining single-cell RNA-Seq and highly multiplex RNA FISH and applied it to human intestinal organoids infected with human astrovirus, a model human enteric virus. We found that interferon controls the infection and that astrovirus infects all major cell types and lineages and induces expression of the cell proliferation marker MKI67. Intriguingly, each intestinal epithelial cell lineage exhibits a unique basal expression of interferon-stimulated genes and, upon astrovirus infection, undergoes an antiviral transcriptional reprogramming by upregulating distinct sets of interferon-stimulated genes. These findings suggest that in the human intestinal epithelium, each cell lineage plays a unique role in resolving virus infection. Our framework is applicable to other organoids and viruses, opening new avenues to unravel roles of individual cell types in viral pathogenesis.

**Keywords** astrovirus; immune response; intestinal epithelial cells; organoids; single-cell transcriptomics
**Subject Categories** Chromatin, Transcription & Genomics; Immunology; Microbiology, Virology & Host Pathogen Interaction
**Mol Syst Biol.** (2021) 17: e9833

## Introduction

The small intestine is responsible for most nutrient absorption in humans. It is composed of various cell types each performing specific functions contributing to homeostasis (Peterson & Artis, 2014). The main cell types found in the intestinal epithelium are the absorptive enterocytes, the mucus-secreting goblet cells, the hormone-producing enteroendocrine cells, the antimicrobial peptide secreting Paneth cells and the stem cells. Due to the constant challenges present in the lumen of the gut, intestinal epithelial cells are turned over every 5 days. This constant self-renewal is organized along the crypt-villus axis and is supported by the stem cells located in the crypts, themselves supported by interlaying Paneth cells (Kretzschmar & Clevers, 2016). Differentiation of the stem cells to the various intestinal cell lineages requires a Notch/Wnt-dependent bifurcation toward either absorptive or secretory progenitor cells. Absorptive progenitor cells give rise to the enterocyte cells while the secretory progenitors differentiate into enteroendocrine, goblet, tuft, or Paneth cells (Sancho *et al*, 2015; Gehart & Clevers, 2019). Over the past 10 years, intestinal organoids have been developed and have emerged as the best surrogate model that mimics the differentiation and function of the intestinal epithelium (Sato *et al*, 2009, 2011).

Human intestinal epithelial cells (hIECs) play key roles in protecting us from environmental pathogen- and commensal-related challenges. They act as a first-layer physical barrier of the host defense and mount response upon infection (Martens *et al*, 2018). Humans are exposed to enteric viruses through contaminated food and water sources as well as through direct fecal–oral transmission from infected patients (Tatte & Gopalkrishna, 2019). People in developing countries are at high risk due to the discharge of

---

1   Structural and Computational Biology Unit, European Molecular Biology Laboratory, Heidelberg, Germany
2   Faculty of Biosciences, Collaboration for Joint PhD degree between EMBL and Heidelberg University, Heidelberg, Germany
3   Department of Infectious Diseases, Molecular Virology, Heidelberg University, Heidelberg, Germany
4   Research Group "Cellular Polarity and Viral Infection", German Cancer Research Center (DKFZ), Heidelberg, Germany
5   Department of Infectious Diseases, Virology, Heidelberg University, Heidelberg, Germany
6   Department of Internal Medicine IV, Interdisciplinary Endoscopy Center, University Hospital Heidelberg, Heidelberg, Germany
7   Flow Cytometry Core Facility, European Molecular Biology Laboratory, Heidelberg, Germany
8   Genomics Core Facility, European Molecular Biology Laboratory, Heidelberg, Germany
9   Molecular Medicine Partnership Unit, European Molecular Biology Laboratory, Heidelberg, Germany
10  Skaggs School of Pharmacy and Pharmaceutical Sciences, University of California San Diego, La Jolla, CA, USA
    *Corresponding author. Tel: +49 6221 56 7865; E-mail: s.boulant@dkfz.de
    **Corresponding author. Tel: +49 6221 387 8690; E-mail: theodore.alexandrov@embl.de
    †These authors contributed equally to this work

---

untreated waste into the environment and lack of medical care after the onset of infection. Diarrheal diseases kill more than 1.5 million people each year worldwide (Tatte & Gopalkrishna, 2019). Depending on the type of virus, infection can lead to gastroenteritis, vomiting, and/or watery diarrhea due to leakage of the intestinal lining. Human astrovirus 1 (HAstV1) is a small non-enveloped, positive-strand RNA virus which is found worldwide (Appleton & Higgins, 1975). These viruses cause a range of symptoms from asymptomatic infections to diarrhea to encephalitis. HAstV1 infections lead to gastroenteritis and account for 2–9% of non-bacterial diarrhea in children (Cortez *et al*, 2017). Most children are exposed to the virus, and by the age of 5 years old, 90% of children have serum antibodies against astrovirus (Walter & Mitchell, 2003; Cortez *et al*, 2017). Human astrovirus infections are often not identified due to lack of diagnostic methods to detect all circulating strains. Work using murine astrovirus in a mouse model of infection has used single-cell approaches to reveal the tropism of murine astrovirus (Cortez *et al*, 2020). It was found that this virus favors goblet cells in the gastrointestinal tract of mice. Interestingly, work using the neurotropic human astrovirus VA1 strain and human intestinal organoids has shown that VA1 has a broader tropism and can potentially infect all major cell types in the gastrointestinal tract (Kolawole *et al*, 2019). However, how individual cell types respond to astrovirus infection was not investigated.

However, only limited knowledge is available about viral pathogenesis in the intestinal epithelium in the context of the cell types, in particular how different human intestinal epithelial cell (hIEC) types contribute to the immune response and clearance of the viral infection. This gap of knowledge is caused by the challenges associated with reproducing the multi-cellular complexity of the human intestinal epithelium. This leads to the current situation where for a majority of enteric viruses, essential questions of viral pathogenesis have been mostly addressed in immortalized cell lines, a model with a limited capacity to reproduce intestinal epithelium in its multicellular complexity and missing key phenomena such as cell differentiation. Interestingly, increasing evidence suggests that stem cells are intrinsically resistant to viral infection (Wolf & Goff, 2009; Belzile *et al*, 2014). It was recently shown that pluripotent and multipotent stem cells exhibit intrinsic expression of interferon-stimulated genes (ISGs) in an interferon-independent manner. This basal expression of ISGs has been proposed to be responsible for the resistance of stem cells to viral infection (Wu *et al*, 2018). Upon differentiation, stem cells lose expression of these intrinsic ISGs and become responsive to interferon (IFN) (Wu *et al*, 2018). These observations suggest that, at least *in vitro*, stem cells and differentiated cells use different strategies to fight viral infection. Whether such cell-type-specific antiviral strategies exist *in-vivo* where stem cells differentiate into different tissue-specific lineages remains unknown. Answering these questions requires using physiologically relevant *ex vivo* models as well as single-cell methods able to dissect the heterogeneity of host–pathogen interactions of different cell types and of individual cells within a population.

Here, we established a framework to investigate cell-type-specific viral pathogenesis in a tissue-like environment. The framework integrates single-cell RNA sequencing and multiplex RNA *in situ* hybridization imaging of enteric virus-infected human ileum-derived organoids. In order to enable cell-type-specific analyzes in organoids, we have created a single-cell RNA-Seq reference dataset

of human ileum biopsies. Applying this framework to organoids infected by human enteric virus astrovirus (HAstV1), we have characterized HAstV1 infection of various primary hIEC types and determined that HAstV1 is able to infect all intestinal cell types with a preference for proliferating cells. Single-cell transcriptomic analysis revealed a cell-type-specific transcriptional pattern of immune response in human intestinal organoids. We found that both at steady state and during viral infection, each intestinal cell type has unique expression profiles of interferon-stimulated genes creating a distinct antiviral environment.

# Results

## HAstV1 infects human intestinal epithelial cells causing IFN-mediated response

To unravel how intestinal epithelium cells respond in a cell-type-specific manner, we investigated human ileum-derived organoids infected with the human pathogen HAstV1. To confirm the ability of HAstV1 to replicate in intestinal cells, we showed that HAstV1 is fully capable of infecting the transformed hIECs, Caco-2 cells, as can be seen by the increase in the number of infected cells and the increase in the amount of viral genome copy number over time (Fig EV1A–C). Following infection with HAstV1, Caco-2 cells mounted an intrinsic immune response characterized by the production of both type I (IFNβ1) and type III (IFNλ) interferons (IFNs) (Fig EV1D). This IFN-mediated response constitutes an antiviral strategy by Caco-2 cells as a pre-treatment of cells with either IFN controlled HAstV1 infection (Fig EV1E and F). This is consistent with previous work reporting that IFN controls HAstV1 infection (Guix *et al*, 2015; Marvin *et al*, 2016). Complementarily, investigating another human intestinal epithelial cell type, T84, which is described to be more immunoresponsive (Stanifer *et al*, 2020a), we found it to be less infectable by HAstV1 unless the IFN-mediated signaling was suppressed by the loss of both the type I and III IFN receptors (dKO) (Fig EV1G). Together, these data confirm the function of IFNs in controlling HAstV1 infection in human intestinal epithelial cells and illustrate the importance of investigating how different cell types mount an interferon response to counteract viral infection.

## HAstV1 infects human ileum-derived organoids

To investigate whether different cell types in the human intestinal epithelium respond to pathogen challenges by mounting a distinct IFN-mediated response, we exploited human intestinal organoids. Human intestinal organoids are an advanced primary cellular system recapitulating the cellular complexity, organization, and function of the human gut and enabling controlled investigations of enteric infection not feasible in the human tissue (Stanifer *et al*, 2020b). Intestinal organoids were prepared from stem cell-containing crypts isolated from human ileum resections (Sato *et al*, 2011). The structural integrity and cellular composition of the organoids were verified by immunofluorescence staining of adherens and tight junctions and markers of various intestinal epithelial cell types (Fig 1A) (Pervolaraki *et al*, 2017). These organoids were readily infectable by HAstV1 as can be seen by the detection of infected

cells and efficient replication of the HAstV1 genome overtime (Fig 1B–E). Infection of organoids by HAstV1 induces a strong intrinsic immune response characterized by the production of both type I IFN (IFNβ1) and type III interferon (IFNλ) at the transcriptional and protein levels (Fig 1F and G). Interestingly, this response was significantly stronger than the one generated by the transformed hIEC lines (Fig EV1), further highlighting the importance of using primary organoids when characterizing host-pathogen interactions. Pre-treatment of organoids with either type I or type III IFN significantly reduced HAstV1 infection (Fig 1H). This strong antiviral activity of IFN against HAstV1 is consistent with the previous reports using both transformed cell lines and human organoids (Kolawole *et al*, 2019) and with the well-described function of IFNs in controlling enteric virus infection at the intestinal epithelium (Lee & Baldridge, 2017).

### Single-cell RNA-Seq profiling of human ileum biopsies

To characterize the response of intestinal organoids to enteric virus infection at the single-cell level, we exploited single-cell RNA sequencing (scRNA-Seq) by using a 10× Genomics platform (Zheng *et al*, 2017). Over the past years, scRNA-Seq emerged as a major approach to reveal cell types, lineages, and their transcriptional programs in tissues. Yet, applying scRNA-Seq to organoids is still challenging due to the available dissociation protocols optimized predominantly for tissues, as well as due to understudied differences between tissues and organoids. In light of these challenges, prior to analysis of ileum-derived organoids, we performed scRNA-Seq analysis of human ileum biopsies with the aim to create a cell-type-annotated reference single-cell dataset of human ileum. Additionally, to increase the resolution of the study, we included in our analysis the recently published scRNA-Seq data of ileum biopsies (Wang *et al*, 2020). Following scRNA-Seq of human ileum biopsies, we quality controlled our scRNA-Seq data (Appendix Fig S1A–I) and we performed data integration of each of our sample and the previously reported datasets (Wang *et al*, 2020) and unsupervised clustering on the resulting integrated space, that revealed the presence of multiple cell subpopulations (Fig 2A). We have identified the cell types represented in the subpopulations by finding subpopulation-specific markers using differential expression of each subpopulation against all other cells (Dataset EV1) and matching them to the known marker genes of the hIEC types (Fig 2B and Appendix Fig S2A). The majority (> 87.5%) of the cells isolated from the biopsies in this study corresponded to hIECs while a small fraction (12.4%) corresponded to stromal and immune cells (Fig 2A and Appendix Fig S2B–D). This integrated dataset represents a detailed reference scRNA-Seq dataset from the human ileum containing 14 cell types (Wang *et al*, 2020) which is especially important for annotating cell types represented in organoids. For all cell types represented in the integrated scRNA-Seq dataset from human ileum biopsies, we selected the most differentially expressed genes as markers (Fig 2B). The pseudotime analysis (Street *et al*, 2018) confirmed the expected bifurcation of differentiation of stem cells along two distinct trajectories toward either enterocytes (absorptive function) or goblet cells (secretory function) (Fig 2C). Each of these two lineages are characterized by specific gradients and waves of gene expression along the differentiation trajectories such as marker genes FCBP and APOA4 (Fig 2D) among others (Fig 2E). This single-cell reference dataset

contains detailed information about transcriptomics profiles of intestinal epithelial cell types in the human ileum that, as we show in the next section, allowed us to annotate multiple cell types in the organoids.

### Human ileum-derived organoids reproduce tissue cell types and lineages

Human ileum organoids were subjected to scRNA-Seq followed by determining the identities of the detected cell types by using cell-type-specific marker genes and label transfer from the ileum biopsies scRNA-Seq data (Fig 3A and Appendix Fig S3). Eleven cell types were identified, including four populations of enterocytes (ALPI, APOA1, FABP6, BEST4), goblet cells (FCGBP, TFF3, SPINK4), enteroendocrine cells (CHGA, CHGB, NEUROD1), stem cells (OLFM4, SMOC2, ASCL2), transient-amplifying (TA) cells (OLFM4, CCL2), cycling TA cells (TUBA1B, MKI67, PCNA), secretory TA cells (SOX4, STMN1), and a subpopulation of stressed TA cells exhibiting high expression of heat shock proteins (HSPA1A, HSPA1B). Most of the anticipated and observed cell types in biopsies were also found in organoids with the exception of Paneth and tuft cells (Fig 3A). This is consistent with previous reports describing the absence of these two intestinal cell types in organoids (Fujii *et al*, 2018) and with the fact that some cell types require specific differentiation protocols to be present in organoids (Rouch *et al*, 2016; Wu *et al*, 2018; Ding *et al*, 2020).

### Multiplex *in situ* RNA hybridization visualizes multi-cellular organization and infection of organoids

ScRNA-Seq is a powerful and robust tool to investigate cell–cell heterogeneity, outline cell types, and determine their marker genes. However, it lacks the capacity for spatial mapping of the discovered information. To overcome this limitation, we employed the HiPlex RNAscope method for multiplexed RNA *in situ* hybridization (Hashikawa *et al*, 2020). This method allows for highly sensitive single-molecule detection of 12 different transcripts, with four transcripts detected simultaneously in each round of hybridization (Appendix Fig S4A). To determine the best cell-type-specific marker genes expressed in organoids, we have repeated the differential analysis and identified marker genes for each individual intestinal cell type; see top markers in Fig 3B and extended list in Dataset EV2. From this list, we selected 11 genes for the multiplexed RNA *in situ* hybridization which were either uniquely expressed in a specific cell type or had less background expression among cells of other types (Appendix Fig S4B). The probes directed against SMOC2, MKI67, CHGB, FCGBP, OLFM4, FABP6, and LYZ showed discrete staining overlapping within cell borders suggesting cell-type specificity. On the other hand, probes directed against SLC2A2, BEST4, LGR5, and SPIB showed either no signal or a background unspecific signal (Appendix Fig S4B). RNA FISH of 11 cell type marker genes, previously not applied at such a high level of multiplexity in organoids (Tsai *et al*, 2017; Seino *et al*, 2018a; Navis *et al*, 2019; Smillie *et al*, 2019), allowed us to validate the expression of these genes and provided a spatial insight into the cell organization in particular with respect to proliferating areas within an organoid (Fig 3C).

To identify which cell types were infected by HAstV1, we designed a HAstV1 RNA-specific probe and performed the

multiplexed RNA *in situ* hybridization on infected organoids. (Fig 3D–I and Appendix Fig S4). *In situ* hybridization of mock-infected organoids using the HAstV1 RNA-specific probe revealed no labeling thus validating the specificity of the HAstV1 probe. In HAstV1-infected organoids, viral RNA was strongly co-localized with the expression of MKI67 (Fig 3D), a marker of proliferation which in our organoid system is expressed predominantly in cycling transit-amplifying cells (Fig 3B). Additionally, mature enterocytes

(marker FABP6) were also found to be infected by HAstV1 (Fig 3E). Viral RNA was also detected, although to lesser extent, in stem-like cells (marker OLFM4) (Fig 3F), enteroendocrine cells (marker CHGB) (Fig 3G), as well as goblet cells (marker FCGBP) (Fig 3H).

For a high-throughput, automated, and quantitative analysis of infected cell types, multiplex *in situ* RNA FISH was performed on 2D seeded organoids. For all RNAscope probes, we considered the respective fluorescence images and calculated single-cell

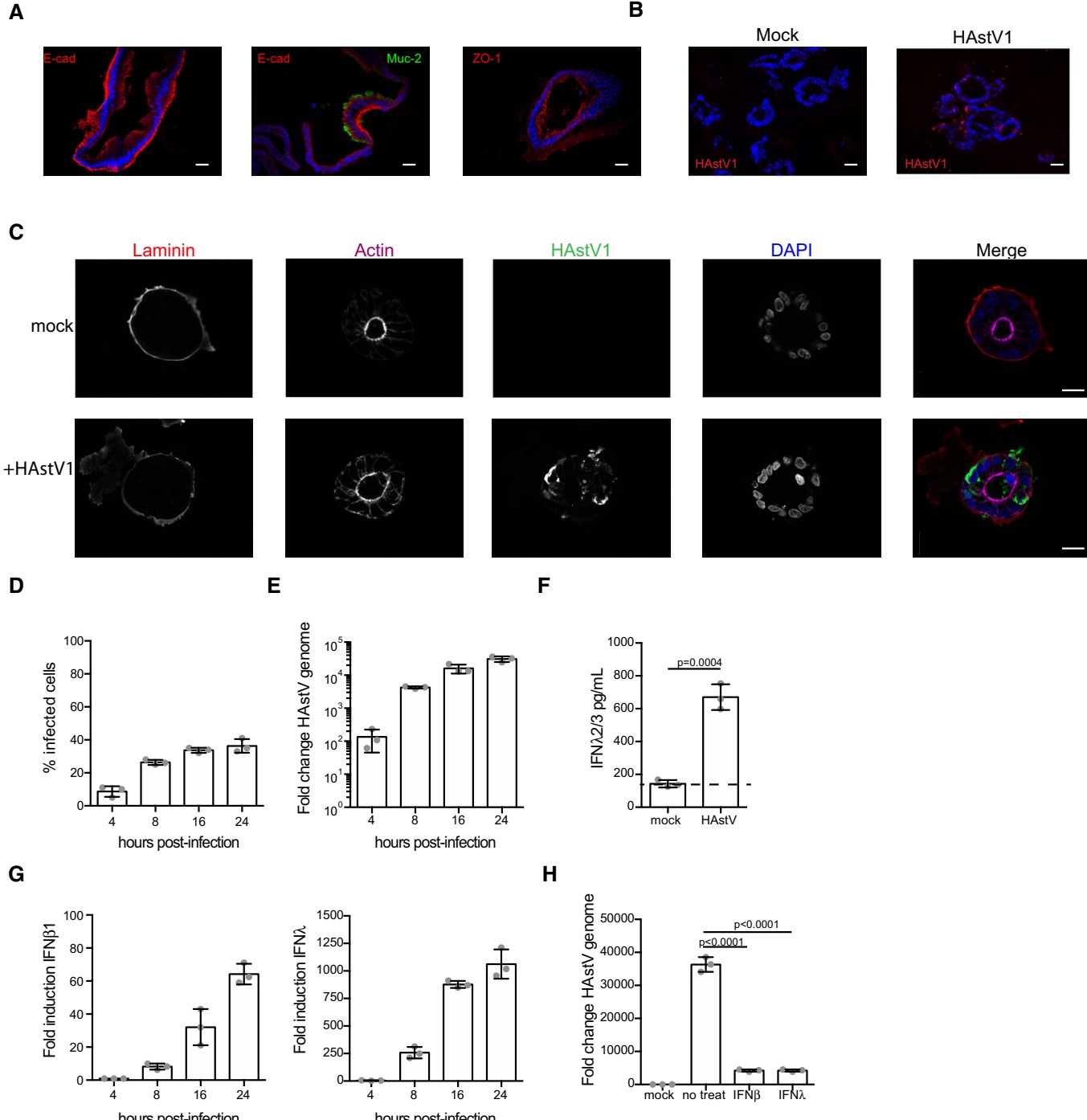

Figure 1.

**Figure 1.   Interferons protect human intestinal organoids from HAstV1 infection.**

A   Cryo-sections of human organoids were analyzed for the presence of enterocytes (E-cad), Goblet cells (Muc-2), and tight junctions (ZO-1) by indirect immunofluorescence. Nuclei are stained with DAPI. Scale bar 25 μm.

B   Human intestinal organoids were incubated with media (mock) or infected with HAstV1. 16 hpi organoids were frozen, cryo-sectioned, and HAstV1-infected cells were visualized by indirect immunofluorescence (HAstV1 (red), nuclei (DAPI, blue). Scale bar 25 μm.

C   Human intestinal organoids were incubated with media (mock) or infected with HAstV1. Organoids at 16 hpi were fixed, and the presence of HAstV1-infected cells (green) was visualized by indirect immunofluorescence. Apical and basolateral membranes were immunostained for actin (magenta) and Laminin (red), respectively. Nuclei are stained with DAPI (blue). Scale bar is 20 μm.

D   Quantification of C with the percentage of infected cells determined.

E   Human intestinal organoids were infected with HAstV1. At indicated time post-infection, the increase in viral copy number was determined by qRT–PCR.

F   Human intestinal organoids were incubated with media (mock) or infected with HAstV1. At 24 hpi, the presence of IFNλ in the media was tested by ELISA. Dotted line indicates detection limit of the assay.

G   Same as E but for the induced levels of either type I IFN (IFNβ1) or type III IFN (IFNλ).

H   Human intestinal organoids were pre-treated for 24 h with 2,000 IU/ml of IFNβ1 or 300 ng/ml of IFNλ1-3. Interferons were maintained during the course of infection and the amount of HAstV1 copy numbers was assayed 24 hpi by qRT–PCR.

Data information: A-G Three biological replicates were performed for each experiment. Representative immunofluorescence images are shown. Error bars indicate the standard deviation. Statistics are from unpaired *t*-test.

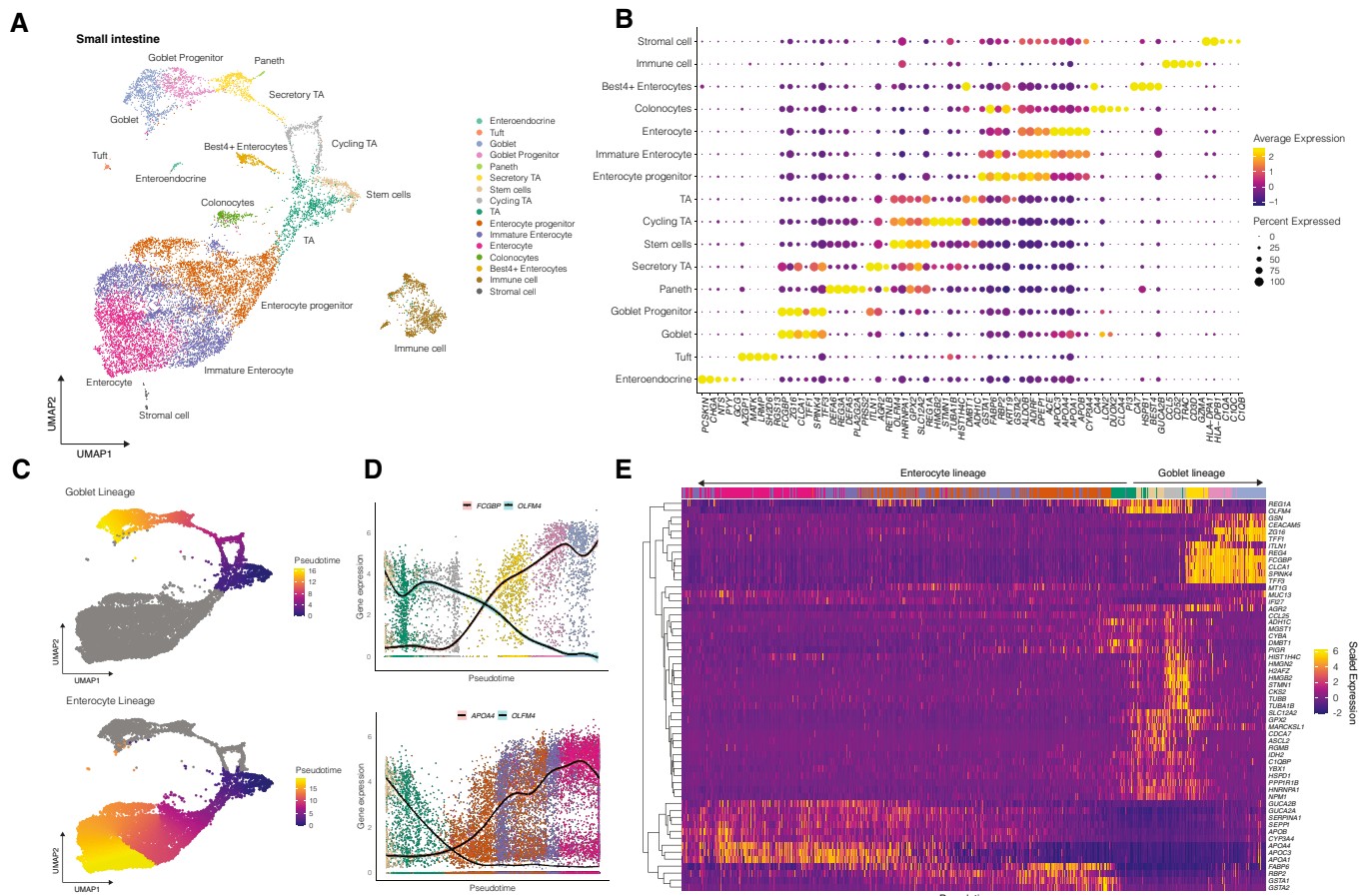

**Figure 2.   Single-cell profiling of human ileum biopsies.**

A   Uniform manifold approximation and projection (UMAP) embedding of single-cell RNA-Seq data from human ileum biopsies colored by the cell type (*n* = 8,800 cells).

B   Dot plot of the top three marker genes for each cell type. The dot size represents the percentage of cells expressing the gene; the color represents the average expression across the cell type. Appendix Fig S3A shows an extended list of the marker genes. All cell-type-defining marker genes are provided in Dataset EV1.

C   Predicted pseudotime trajectories for the goblet (top) and enterocyte lineages (bottom) projected onto the UMAP embedding.

D   Normalized gene expression for genes associated with differentiation of the goblet (top) and the enterocyte lineage (bottom)

E   Heatmap showing changes in gene expression levels along pseudotime for the enterocytes and goblet lineages. Dendrograms on the left of the heatmap indicate the results of hierarchical clustering of the genes. Colored bars above the heatmap indicate the types of cells ordered along the pseudotime, colored according to A.

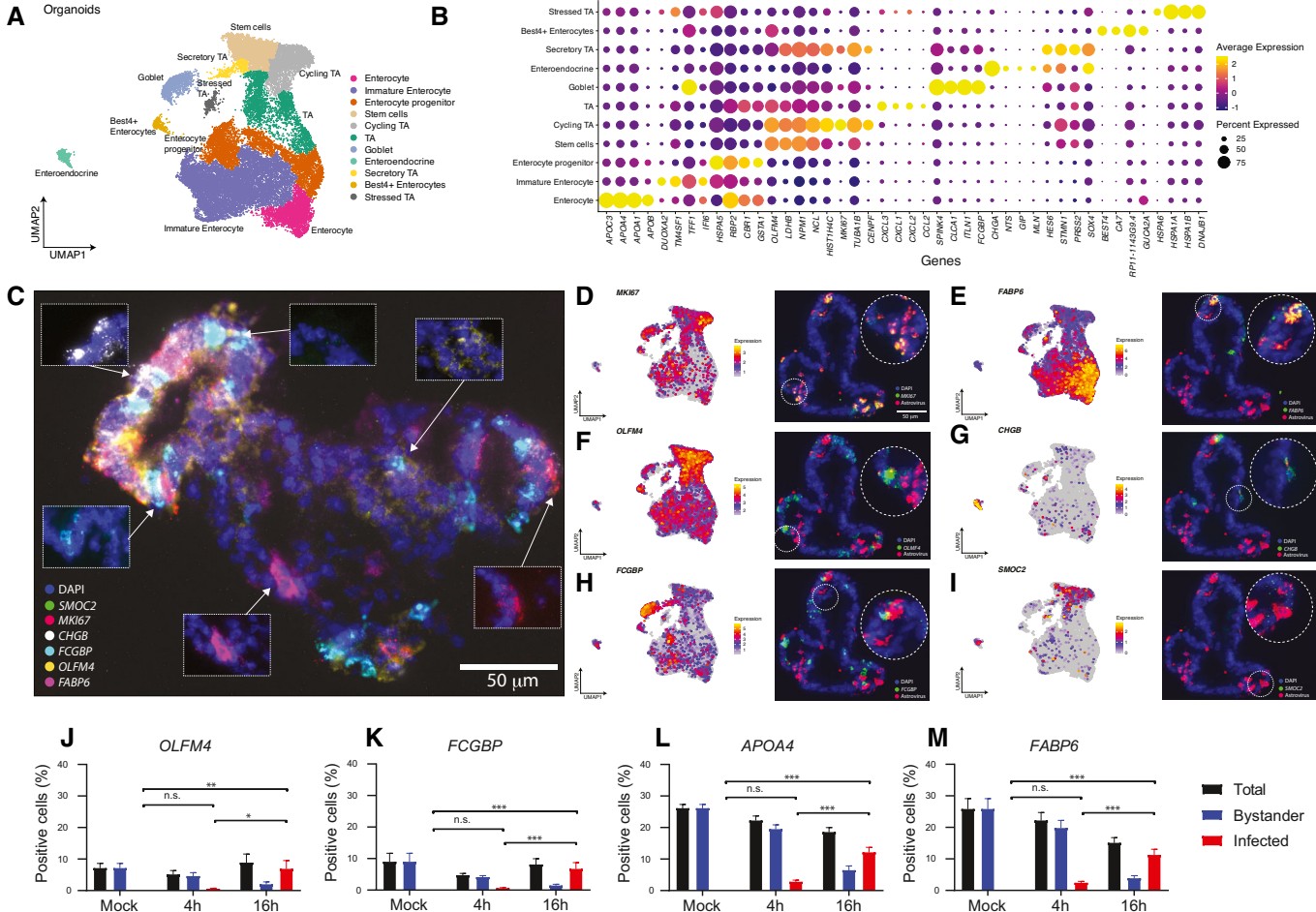

**Figure 3. Single-cell profiling and multiplex *in situ* RNA hybridization of human ileum-derived organoids.**

A    UMAP of scRNA-Seq data from human ileum-derived organoids (*n* = 16,682 cells); dots corresponding to cells are colored by the cell type.
B    Dot plot of the top three marker genes for each cell type. The dot size represents the percentage of cells expressing the gene; the color represents average expression across the cell type. Appendix Fig S3B shows an extended list of the marker genes. All cell-type-defining marker genes are provided in Dataset EV2.
C    Representative images showing multiplex *in situ* RNA hybridization of marker genes in an organoid. DAPI in blue.
D–I    Visualizations for cell type markers for the major intestinal cell types, showing single-cell expression intensities of the cell type markers on the UMAP (left) and multiplex *in situ* RNA hybridization; red corresponds to viral RNA and a green corresponds to cell type markers (right).
J–M    Percentage of stem cells (OLFM4) goblet cell (FGCBP), mature enterocytes (APOA4), and cells of the enterocyte lineage (FGCBP) from the total organoid cells population (black bar) in mock-treated organoids and after HAstV1 infection. The percentage of infected cells (red bars) and bystander cells (blue bars) within each cell type is shown. The data were obtained from multiplex in situ RNA hybridization in 2D organoids. *n* = 10, mean ± SEM. Statistics show comparison of infected cells between mock-treated, 4 and 16 hpi organoids. An ordinary one-way ANOVA and Tukey's multiple comparisons test was used. n.s nonsignificant, \**P* < 0.05, \*\**P* < 0.01, \*\*\**P* < 0.001.

Source data are available online for this figure.

fluorescence intensity values which represent RNA expression levels for the considered genes. This included applying background correction, nuclei segmentation, fluorescence measurement, and elimination of false-positive and false-negative signals. Using 2D organoids helped ensure high quality of microscopy since all organoid cells are in one focal plane. This allowed us to quantify the expression of genes from the RNAscope panel for several thousand single cells in an automated, quantitative and reproducible manner. For four representative marker genes, a threshold was set to determine positive and negative cells for each cell type marker, that allowed us to estimate the percentage of each cell type in the total population. As

expected, stem cells (OLFM4) comprise < 7% of the total organoid cell population at steady state (mock-treated organoids) (Fig 3J). Similarly, goblet cells (FCGBP) represent a small part of the organoid cells (< 10%) (Fig 3K) while around 25% of the cells are positive for the enterocyte markers (APOA4 and FABP6) (Fig 3L and M). To identify infected cells, the fluorescent signal from the HAstV1 RNA-specific probe was used. While the infection was already visible at 4 h post-infection (hpi), the percentage of infected cells at 16 hpi was significantly higher for all cell types when compared to mock-treated organoids (Fig 3J–M). Surprisingly, HAstV infection also seems to reduce the total amount of

enterocytes (Fig 3L and M). While 25% of mock-treated organoids are FABP6-positive, at 16 hpi only 15% of cells are positive for the enterocyte lineage marker (Fig 3M). Whether enterocytes lose the lineage specification upon infection or the cell type marker fluorescence is below the detection limit needs further investigation. Altogether, these results indicate the potential of HAstV1 to infect most hIECs and are in agreement with a recent report showing that HAstV1 can infect most human intestinal epithelial cell types (Kolawole *et al*, 2019).

Finally, our multiplex *in situ* RNA FISH analysis performed in 3D organoids shows that the proliferation marker MKI67 strongly colocalizes with the viral RNA (Fig 3D). Therefore, we quantified the phenotype using the automated, high-throughput, and quantitative analysis applied to RNA FISH images from 2D organoids as described in the previous paragraph. Representative images show that, while MKI67 is not expressed in mock-infected organoids, at 16 hpi most infected cells are positive for MKI67 (Fig EV2A). Accordingly, at the single-cell level the MKI67 expression strongly positively correlates ($R^2 = 0.8347$) with HAstV1 RNA (Fig EV2B). Furthermore, mock-treated organoids have very low MKI67 expression levels, which is only slightly increased at the early time point of 4 hpi (Fig EV2B). We also see an induction of the cell cycle marker over infection time when analyzing every cell type individually in particular in the enterocyte lineage (FABP6 and APOA4) (Fig EV2C). These results strongly suggest that HAstV1 infection induces the expression of the cell proliferation marker MKI67.

### HAstV1 evokes cell-type-specific transcriptional response

To address whether the cell types present in the intestinal organoids differently respond to viral infection, we performed scRNA-Seq of HAstV1-infected organoids. Single-cell transcriptional analysis confirmed that all intestinal epithelial cell types could be infected by HAstV1 (Fig 4A and B). Additionally, all cell types support virus replication with different levels of viral expression present in each cell type at 16 hpi (Appendix Fig S5). Differential gene expression analysis of mock-infected organoids versus 4 and 16 hpi with HAstV1 revealed upregulation of multiple pathways upon infection, including the pathways linked to infection, pro-inflammatory response, and interferon-mediated response (Fig EV3 and EV4). Interestingly, differential gene expression analysis integrating the different cell types revealed a cell-type-specific transcriptional response upon virus infection (Fig 4C–E). This response includes canonical marker genes for the detected cell types, *e.g.*, upregulation of APOA1 in enterocytes, downregulation of OLFM4 in stem cells and TA cells, and upregulation of CHGA in enteroendocrine cells (Fig 4E). Most importantly, our analysis revealed that different ISGs are upregulated in different cell types upon viral infection (Fig 4C and D, expanded description for enterocytes, enteroendocrine, and TA cells in Fig EV3 and for stem cell, goblet cells, and best+4 cells). These findings suggest that each cell type present in our intestinal organoids has a unique transcriptional program to combat viral infection.

To better evaluate the involvement of individual cell types into the interferon-mediated response to HAstV1 infection (Fig 1), we performed detailed differential analysis using ISGs only (Fig 4F). For the enterocytes lineage, 68 out of 80 ISGs were found to significantly change either at 4 or 16 hpi compared to mock with almost all significant ISGs upregulated at 16 hpi. Other cell types including

stem and TA cells demonstrated their cell-type-specific changes at both early (4 hpi) and late (16 hpi) time points (Fig 4F).

Surprisingly, the unsupervised mapping of all cells based on their ISG profiles revealed an organization of cells according to the cellular lineage (Figs 4G and EV5B and C). Further clustering of cells based on their ISG profiles revealed four clusters (Figs 4H and EV5E and F) with cluster 1 being mostly composed of stem and TA cells and cluster 4 being mostly composed of enterocytes (Figs 4I and EV5H and I). Differential analysis revealed distinct patterns of ISGs expressed upon HastV1 infection which are specific to the cell types represented in the outlined clusters (Figs 4J and EV5K and L). Importantly, similar analyses of mock-infected organoids also revealed a lineage-specific clustering of cells according to their ISG expression profiles (Fig EV5A, D, G and J). Even in mock-infected organoids, each cluster expresses different basal levels of distinct ISGs (Fig EV5J). Altogether, these findings strongly support that each intestinal cell lineage is characterized by a distinct basal signature of ISG expression and that upon viral infection, each intestinal cell lineage responds differently by mounting a distinct immune response characterized by cell-type-specific patterns of ISG expression.

### Multiplex *in situ* RNA hybridization visualizes cell-type-specific immune signature in HastV1-infected and bystander cells

To validate the unique immune signature of each cell lineage in the presence or absence of HastV1 infection, expression of immune-signaling associated genes was quantified using multiplex RNA *in situ* hybridization of genes determined to be highly expressed in either cluster 1 or cluster 4 (Fig 4J). Representative images show expression of CCL2 in stem cells (OLFM4-positive cells), goblet cells (FGCBP-positive cells), and enterocytes (FABP6- and APOA4-positive cells) for mock-treated and infected organoids (Fig 5A and Appendix Fig S6). In mock-treated organoids, most of the cells are negative for CCL2 signal, while expression is induced upon viral infection especially at 4 hpi (Fig 5A and Appendix Fig S6). Interestingly, a high correlation between the stem cells (OLFM4) and the goblet cells (FCGBP) markers and CCL2 was observed as early as at 4 hpi (arrows) (Appendix Fig S6A and B). Although upregulation of CCL2 expression in these two cell types is mostly seen in infected cells (white arrows), bystander cells also express the immune-signaling gene (yellow arrows). On the contrary, CCL2 is more often present in cells that are negative for enterocyte markers FABP6 (Fig 5A) and APOA4 (Appendix Fig S6). As a side note, one can see the FABP6 and APOA4 fluorescent signals to be reduced at 16 hpi when compared to mock-treated organoids. This indicates that infection affects the enterocyte lineage in organoids. To quantitatively evaluate these differences in gene expression, the fluorescence intensity of the ISGs CCL2, IFNGR1, IRF2, IFI27, and STAT1 were determined on a single-cell level for the different cell lineages in either mock or HAstV1-infected organoids. Infected and bystander cells were plotted separately to analyze whether infection induces a specific immune signature. Consistent with previous results, the upregulation of STAT1 expression over time shows that HAstV1 infection in organoids is linked to an IFN-mediated response (Fig 5C). Furthermore, when analyzing the overall ISGs induction, it is clear that infected cells elicit a faster and stronger upregulation of ISGs compared to bystander cells, which is especially noticeable at 4 hpi (Fig 5B–D and Appendix Fig S7).

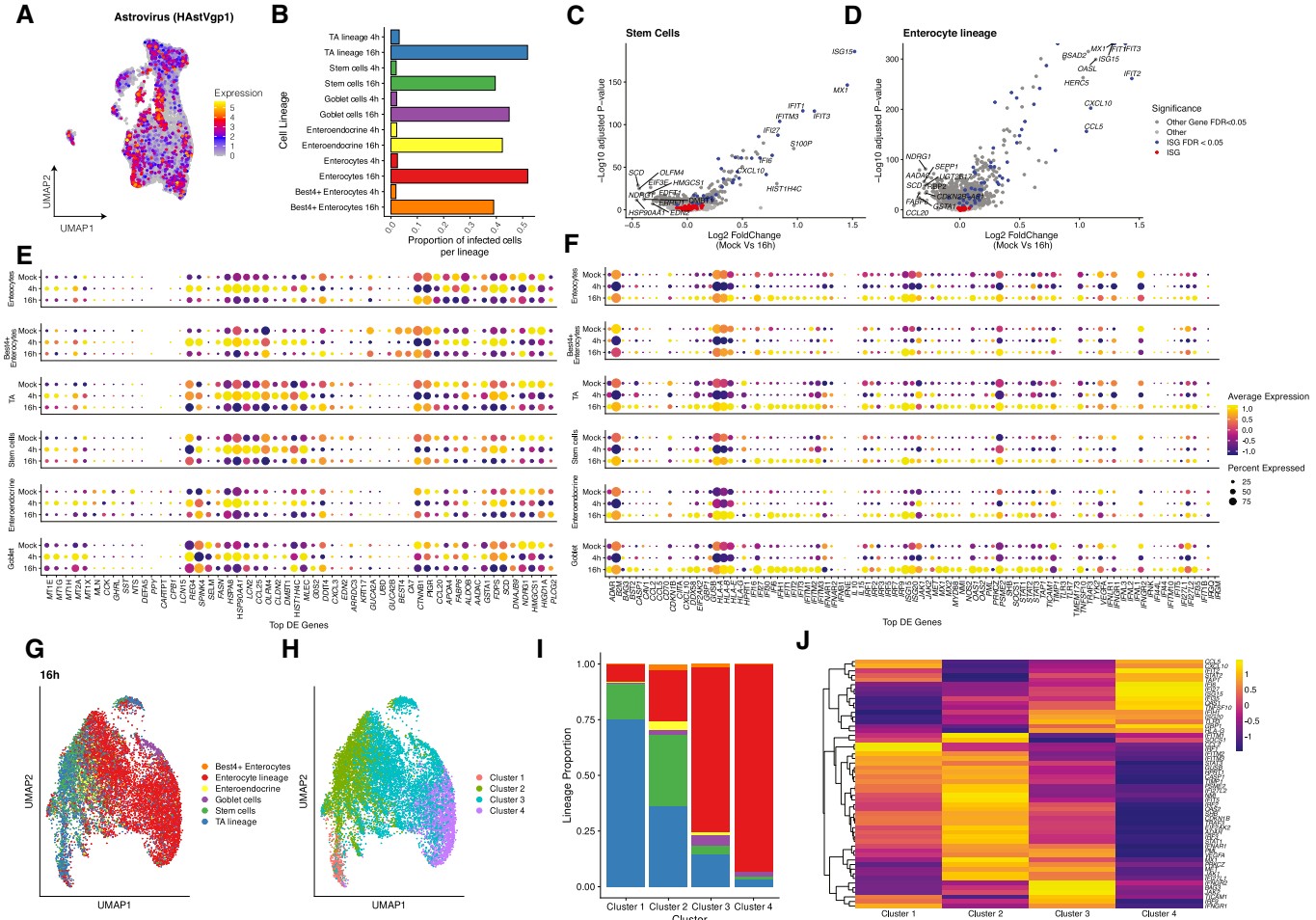

**Figure 4. Cell-type-specific ISG induction upon HAstV1 infection of human ileum organoids.**

A   UMAP embedding of human ileum-derived organoids colored by the expression of HAstV1.

B   Proportion of cells infected with astrovirus detected in each cell lineage and type.

C, D  Volcano plots of genes that are differentially expressed in mock relative to 16 hpi, showing the statistical significance (−log10 adjusted *P*-value) versus log2 fold change (mock/16 hpi) for the stem cells (C) and enterocyte lineage (D).

E   Dot plot of top lineage-specific expression changes upon HAstV1 infection for mock, 4 and 16 hpi organoids.

F   Same as in E but for ISG. The dot size represents the percentage of cells expressing the gene; the color represents the average expression across the lineage. Triangle and stars represent significantly changing genes (FDR < 0.05) in mock-infected samples versus 4 hpi as well as and mock-infected samples versus 16 hpi, respectively.

G   UMAP embedding of scRNA-Seq data from human ileum-derived organoids based on the significantly changing ISGs for cells at 16 hpi.

H   Unsupervised clustering of the UMAP data from G.

I   Distribution of cell lineages and types in the clusters from H.

J   Heatmap of differentially expressed ISGs across the clusters from H.

When addressing the expression of ISGs in individual cell lineages, we noticed differences in their basal expression levels in mock-treated organoids (Fig 5B–D and Appendix Fig S7). Interestingly, stem cells seem to have significantly higher basal ISG expression than the goblet cells and enterocytes. Our multiplex RNA *in situ* hybridization data confirmed our scRNA-Seq cells clustering (Fig 4H–J) showing that stem and goblet cells have higher levels of CCL2 (Fig 5B), STAT1 (Fig 5C), IFNGR1 (Fig 5D), and IRF2 (Appendix Fig S7A) than enterocytes, especially at 4 hpi. At the same time, IFI27 expression levels appear to be higher in the enterocyte lineage when compared to levels in stem and goblet cells at 16

hpi (Appendix Fig S7B and Fig 4H–J). These results further highlight that each cell lineage responds differently to HAstV1 infection by inducing a characteristic activation of ISGs. Furthermore, we could determine the immune signature in infected versus bystander cells and showed that infected cells induce a stronger antiviral immune response than bystander cells. Altogether, our data show that HAstV1 can infect most cell lineages present in the human intestinal epithelium and that each lineage generates a distinct immune response upon infection. Importantly, this lineage-specific expression of ISGs does not appear to result from virus infection since mock-infected cells also display lineage-specific expression of ISGs.

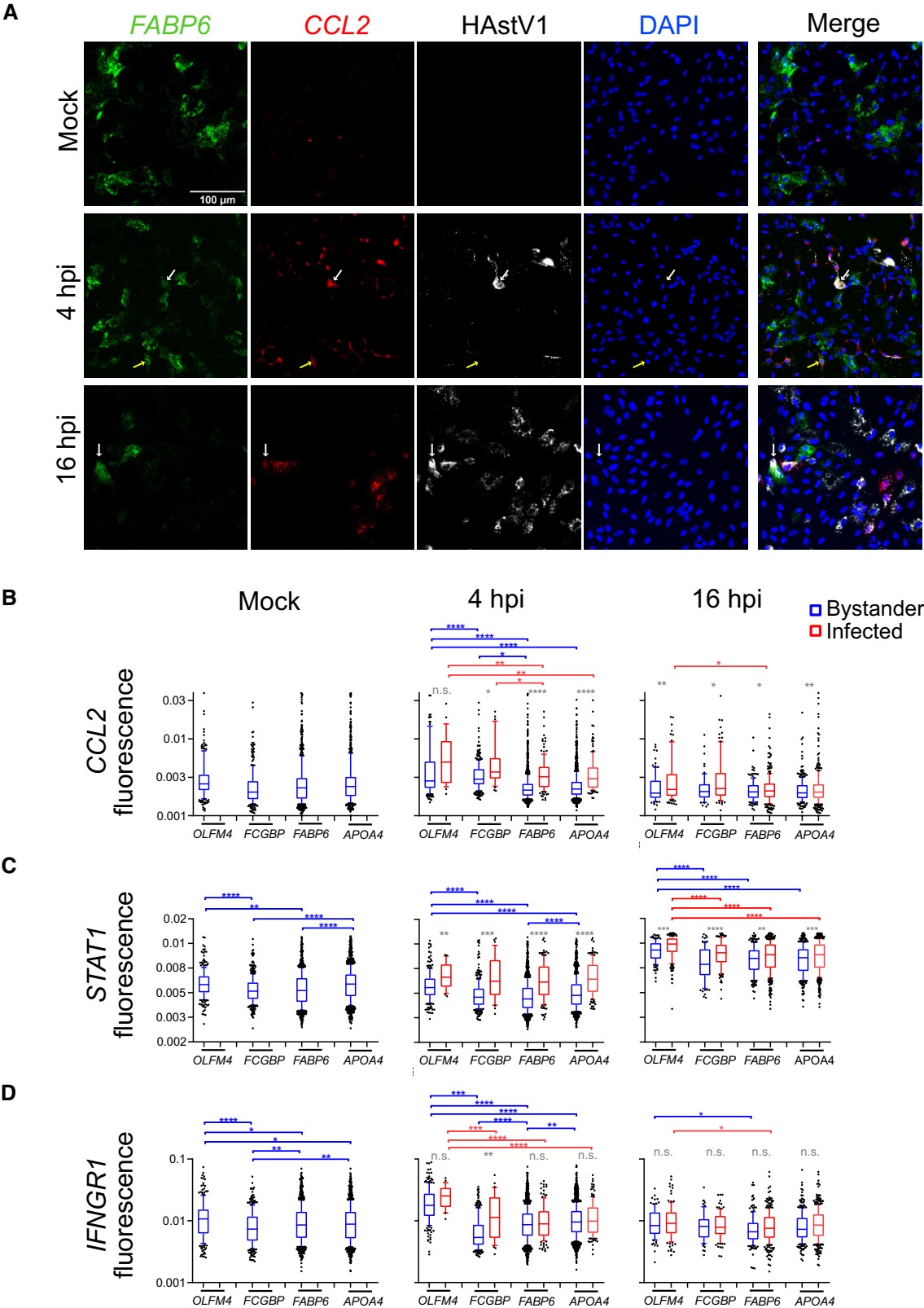

**Figure 5.**

**Figure 5.  Multiplex *in situ* RNA FISH visualizes immune signature in HastV1-infected and bystander cells.**

A       One image per condition was used to show the multiplex in situ RNA FISH. A representative region of the images was chosen and cropped in order to have a zoom-in of a specific area for better visualization. Shown are the enterocyte lineage marker FABP6 (green), HAstV1 infection (white), and CCL2 gene expression (red) of mock-treated and infected organoids. DAPI is in blue. White arrows show co-localization of FABP6 and CCL2 with HAstV1 signal. Scale bar 100 μm.

B–D    Fluorescence intensity (arbitrary units) of innate immune marker expression in stem cells (OLFM4 positive), goblet cells (FCGBP positive), enterocyte lineage cells (FABP6 positive), and mature enterocytes (APOA4-positive). Bystander cells are in blue and infected cells in red. 10–90 percentile box plots, each dot represents one cell (mock $N = 2,274$, 4 h $N = 3,682$, 16 h $N = 1,510$. (C) mock $N = 2,693$, 4 h $N = 3,706$, 16 h $N = 1,921$. (D) mock $N = 2,274$, 4 h $N = 3,706$, 16 h $N = 1,514$). Blue statistics show a comparison of bystander cells between cell lineages, and red statistics show the comparison of infected cells between cell lineages. Ordinary one-way ANOVA and Tukey's multiple comparisons test were used. Gray statistics show the comparison between infected and bystander cells within each cell lineage. Unpaired *t*-test with Welch's correction was used. n.s nonsignificant, $*P < 0.05$, $**P < 0.01$, $***P < 0.001$, $****P < 0.0001$.

Source data are available online for this figure.

## Discussion

In this work, we exploited human intestinal organoids, scRNA-Seq, and multiplex RNA *in situ* hybridization to characterize the response to infection by the enteric virus HAstV1 in a primary model system that recapitulates the cellular complexity of the human gut. Using multiplex RNA *in situ* hybridization spatially correlating viral infection to cell types of an organoid and scRNA-Seq, we found that HAstV1 infects all detected cell types and induces expression of the cell proliferation marker MKI67. Importantly, it revealed that each individual intestinal cell lineage mounted a distinct transcriptional response upon viral infection. We found cell-type-specific transcriptional patterns of ISGs expression, with different lineages characterized by the expression of distinct ISGs. Importantly, our analysis also revealed that the basal level of ISG expression in non-infected organoids was also cell-type-specific. These findings strongly suggest that within the intestinal epithelium, each individual cell type utilizes a different strategy to combat viral infection and has a different function in the clearance of the pathogen.

Investigating host–pathogen interactions and viral pathogenesis in tissue is challenging. It requires identifying cell types composing the tissue and correlating virus infection to each cell type within its spatial context. Here, we have developed a framework combining scRNA-Seq and multiplex RNA *in situ* hybridization to overcome these limitations. Associating transcriptional changes to specific cell types in scRNA-Seq data requires validated cell-type-specific marker genes. Despite the availability of single-cell atlases for other parts of the intestine published recently (Smillie *et al*, 2019), no reference atlas was available for the small intestine when we started this work. Recently, Wang *et al* (2020) published a scRNA-Seq dataset of human ileum with a total of 6,167 epithelial cells and 7 cell types annotated. Our scRNA-Seq dataset from human ileum biopsies is in line with the data published by Wang *et al* (2020), and both datasets were used for creating a reference dataset and annotating cell types. Having a comprehensive coverage of cell types and differentiation lineages in a reference dataset is especially important when working with organoids due to possible transcriptional changes between organoid and tissue cells as well as the presence of not fully differentiated cells that otherwise may lead to false annotations.

Applying scRNA-Seq to study infection in multi-cellular models or tissue is still challenging as it is a sophisticated technology requiring optimization of cell dissociation, fixation, and pathogen inactivation. This leads to the current situation when scRNA-Seq is applied mainly to study viral infection in cell lines (O'Neal *et al*,

2019; Wyler *et al*, 2019; Sun *et al*, 2020) or primary tissue where one cell type is present or only mature cells are present like PBMCs (Zanini *et al*, 2018) or lung epithelial cells (preprint: Cao *et al*, 2020). Optimizing our framework, we have evaluated whether fixation or sorting should be performed and concluded that using live cells without fixation provided the best quality scRNA-Seq data for infected organoids (Appendix Fig S8A–E) and that using sorting introduces no significant differences (Appendix Fig S8F–J).

Although scRNA-Seq provides a comprehensive view of transcriptional programs and serves as an excellent tool for outlining cell types, subtypes, and differentiation lineages, we found RNA *in situ* hybridization to be crucially important as an orthogonal way to provide spatial information about the expression of marker genes. Using *in situ* hybridization for the viral RNA represents a native approach to integrate information about cell types and infection. First, it can be more sensitive and specific for virus localization compared to immunofluorescence, and importantly, it represents a fast alternative in case there is no specific antibody available. Second, this approach allowed us to bypass the main limitation of correlating scRNA-Seq with fluorescence microscopy which is often limited by the availability of antibodies for the different cell type markers. Most importantly, using scRNA-Seq and RNA *in situ* hybridization together for cell type analysis leads to further advantages compared to complementing scRNA-Seq with detecting cell types using antibodies. A limited correlation between abundances of RNA molecules and cognate proteins can lead to discrepancies of the cell type definitions and/or corresponding cell type markers between the transcript and proteins levels. Localizing viral RNA together with cell-type-specific marker genes requires multiplexing. Here, we show the first application of highly multiplex (> 10 channels) RNA *in situ* hybridization to infected organoids whereas previous studies reported detection of one or two cell types (Tsai *et al*, 2017; Seino *et al*, 2018b; Navis *et al*, 2019; Smillie *et al*, 2019). We anticipate our framework for integrative single-cell RNA sequencing and RNA FISH of organoids to be widely applicable to other organoid models and to any RNA virus with poly(A) tails thus representing a novel approach to study viral pathogenesis and able to provide novel insights into immune response and cell-type-specific transcriptional reprogramming upon infection.

Understanding enteric virus pathogenesis and how the intestinal epithelium combats viral infection has become a growing field of interest due to the socioeconomic impact of viral gastroenteritis (Bányai *et al*, 2018). Interferons and the associated interferon-stimulated genes are well known to contribute to the first-line

defense against viruses. Among the different types of IFNs, type III IFN (*i.e.*, IFNλ) (Kotenko *et al*, 2003; Sheppard *et al*, 2003) has been shown to be the key antiviral cytokine controlling enteric viruses (Pott *et al*, 2011; Nice *et al*, 2015; Pervolaraki *et al*, 2017, 2018). Our scRNA-Seq analysis confirms that most HAstV1-infected cells show high expression of IFNλ and ISGs. On the contrary, type I IFN (IFNβ1) was not detected in our scRNA-Seq data, confirming the type III IFN preference of IECs in response to viral infection (Mahla-kõiv *et al*, 2015).

Recently, scRNA-Seq has also been used to investigate murine astrovirus infection of the murine intestinal tract (Cortez *et al*, 2020). In this study, the authors found that murine astrovirus prefers to infect goblet cells and that mucus secretion is impacted upon infection (Cortez *et al*, 2020). This limited tropism of murine astrovirus is different from what has been reported for human astroviruses. Immunohistochemistry staining of human biopsies has shown that HAstV1 prefers to replicate in the enterocytes while immunofluorescence and FACS staining have shown that the human VA1 strain has a broader cell tropism infecting most cell types (Sebire *et al*, 2004; Kolawole *et al*, 2019). Using both scRNA-Seq and multiplex RNA scope, we can detect virus replication in all cell types where around 40% of cells become infected with HAstV1 which is further corroborated by our immunofluorescence staining. We showed that this is an active replication as we see an increase of the virus genome from 4 to 16 hpi and can detect high levels of genome copies in all cell types (Fig 4A and B, Appendix Fig S5). These results highlight how murine and human astrovirus can display a distinct tropism for the same tissue and show that further studies are required to determine if these differences in tropism are due to receptor accessibility or the presence of cell-type-specific antiviral factors. Importantly, although both us and (Kolawole *et al*, 2019) used intestinal organoids which are the best surrogate model to mimic the human gut, organoids have their limitations. It is possible that the extended tropism observed in the organoid models is a result of the simplistic nature of the organoid, e.g., due to the absence of microbiota and tissue-resident immune cells.

Stem cells have been known to be highly resistant to viral infection compared to differentiated cells. It has been shown that although these cells are refractory to IFNs, they express a basal level of ISGs which are responsible for their intrinsic resistance to viral infection (Wu *et al*, 2018). Interestingly, different tissue stem cells show a distinct subset of ISG expression patterns. Differentiation of human embryonic stem cells to endo-, meso-, and ectoderm also shows that the three different lineages have a different basal level of ISGs (Wu *et al*, 2018). Here, by using intestinal organoids, we could show that in a naturally differentiating system all intestinal lineages and their progenitor cells display distinct basal expression levels of ISGs. This may contribute to viral tropism by creating an antiviral state restricting virus in a cell-type-specific manner.

The use of intestinal organoids to study host–enteric pathogen interaction is only at its premise (Dutta & Clevers, 2017). Bulk transcriptional analysis of viral infection in intestinal organoids, which are fully immunoresponsive compared to many intestinal-derived cell lines, shows that they induce a typical IFN-mediated antiviral response (Kolawole *et al*, 2019; Lamers *et al*, 2020). ScRNA-Seq of both primary and immortalized virus-infected cell lines revealed a population heterogeneity of both viral replication and the

transcriptional response of cells to viral infection (Combe *et al*, 2015; Wyler *et al*, 2019; Russell *et al*, 2018). Here, we report the first scRNA-Seq study of viral infection of organoids. Our data not only show an IFN-mediated antiviral response but also highlight how individual intestinal lineages contribute to mounting the overall response by activating unique transcriptional programs characterized by cell-type-specific subsets of ISGs (Figs 4 and EV5). Interestingly, we also observed a heterogeneity of the transcriptional response upon infection within cells of the same lineage. However and most importantly, clustering cells based on their ISGs profiles reveal that the differences of the ISG expression between intestinal cell lineages is higher than the heterogeneity between cells of the same lineage (Fig EV5). This cell type specificity of expression of ISGs observed for the enterocyte lineage, stem cells, transit-amplifying cells, and goblet cells confirms the potential of using bioengineered tissue-like systems to dissect the cell-type-specific immune response upon host–pathogen interactions. From a biological point of view, our findings strongly suggest that individual cell types within the intestinal epithelium exert different functions during pathogenesis and contribute differently to mounting the immune response and to the overall clearance of the pathogen.

# Materials and Methods

### Cells and viruses

T84 human colon carcinoma cells (ATCC CCL-248) were maintained in a 50:50 mixture of Dulbecco's modified Eagle's medium (DMEM) and F12 (GibCo) supplemented with 10% fetal bovine serum and 1% penicillin/streptomycin (Gibco). Caco-2 human colorectal adenocarcinoma (ATCC HTB-37) was maintained in Dulbecco's modified Eagle's medium (DMEM) (GibCo) supplemented with 10% fetal bovine serum and 1% penicillin/streptomycin (Gibco). Human Astrovirus 1 (HAstV1) was a kind gift from Stacy Schultz-Cherry, St. Jude Children's Research Hospital, TN, USA. HAstV1 was amplified in Caco-2 cells and virus-containing supernatants were used for infection experiments. P3 stocks of the virus were used for all experiments. MOI was determined by a TCID50 assay on Caco-2 cells.

### Human organoid cultures

Human tissue was received from colon and small intestine resection from the University Hospital Heidelberg. This study was carried out in accordance with the recommendations of the University hospital Heidelberg with written informed consent from all subjects in accordance with the Declaration of Helsinki. All samples were received and maintained in an anonymized manner. The protocol was approved by the "Ethics commission of the University Hospital Heidelberg" under the protocol S-443/2017. Stem cells containing crypts were isolated following 2 mM EDTA dissociation of tissue samples for 1 h at 4°C. Crypts were spun and washed in ice-cold PBS. Fractions enriched in crypts were filtered with 70 μM filters, and the fractions were observed under a light microscope. Fractions containing the highest number of crypts were pooled and spun again. The supernatant was removed, and crypts were resuspended in Matrigel. Crypts were passaged and maintained in basal and differentiation culture media (see Appendix Table S1).

## Antibodies and inhibitors

ZO-1 (Santa Cruz Biotechnology) was used at 1/100 for immunostaining; Phalloidin-AF647 (Molecular Probes) was used at 1/200 for immunostaining; Laminin (Abcam) was used at 1/100 for immunostaining, mouse monoclonal against E-cadherin (BD Transductions), and rabbit polyclonal anti-Mucin-2 (Santa Cruz Biotechnology) were used at 1:500 for immunofluorescence, and HAstv1 capsid antibody (Abcam) was used at 1/300 for immunostaining. Secondary antibodies were conjugated with AF488 (Molecular Probes), AF568 (Molecular Probes), and AF647 (Molecular Probes) directed against the animal source.

## RNA isolation, cDNA, and qPCR

RNA was harvested from cells using NuceloSpin RNA extraction kit (Machery-Nagel) as per manufacturer's instructions. cDNA was made using iSCRIPT reverse transcriptase (Bio-Rad) from 250 ng of total RNA as per manufacturer's instructions. qRT–PCR was performed using iTaq SYBR green (Bio-Rad) as per manufacturer's instructions, HPRT1 was used as normalizing genes. Primer used is described in Appendix Table S2.

## Indirect immunofluorescence assay

Cells seeded on iBIDI glass bottom 8-well chamber slides. At indicated times post-infection, cells were fixed in 4% paraformaldehyde (PFA) for 20 min at room temperature (RT). Cells were washed and permeabilized in 0.5% Triton-X for 15 min at RT. Primary antibodies were diluted in phosphate-buffered saline (PBS) and incubated for 1 h at RT. Cells were washed in 1X PBS three times and incubated with secondary antibodies and DAPI for 45 min at RT. Cells were washed in 1× PBS three times and maintained in PBS. Cells were imaged by epifluorescence on a Nikon Eclipse Ti-S (Nikon).

Organoids were fixed in 2% paraformaldehyde (PFA) for 20 min at room temperature (RT). Cells were washed and permeabilized in 0.5% Triton-X for 15 min at RT. Primary antibodies were diluted in phosphate-buffered saline (PBS) and incubated for 1 h at RT. Antibody was removed, and samples were washed in 1× PBS three times and incubated with secondary antibodies for 45 min at RT. Antibody was removed, and samples were washed in 1× PBS three times and kept in water for imaging. Organoids were imaged on an inverted spinning disk confocal microscope (Nikon, PerkinElmer) with 60× (1.2 numerical aperture, PlanApo, Nikon) water immersion objective and an EMCCD camera (Hamamatsu C9100-23B).

## ELISA

Supernatants were collected at 16 h post-infection. Supernatants were kept undiluted. IFNλ2/3 was evaluated using the IFNλ2/3 DIY ELISA (PBL Interferon source) as per manufacturer's instructions.

## Organoid infection

Organoids were moved to differentiation media (Appendix Table S1) 3 days prior to infection. At the time of infection, medium was removed and organoids were resuspended in cold PBS to remove any excess of Matrigel. To allow for infection from both the apical and basolateral sides, organoids were disrupted with a 27G needle 10 times. Mock infection cells were incubated with media, and infected samples were incubated with supernatant containing virus. Infections were allowed to proceed for 1 h. Cells were then spun down and virus was removed. Cells were washed one time with DMEM/F12 and then resuspended in differentiation media for the course of the experiment.

## Tissue and organoid dissociation for scRNA-Seq

Obtained fresh human ileal biopsies were kept in PBS at 4°C until processing using the previously described method (Smillie *et al*, 2019). The biopsy material was cut into small pieces using scissors. The dissociated tissue was then incubated in dissociation solution (HBSS Ca/Mg-Free, 10 mM EDTA, 100 U/ml penicillin, 100 mg/ml streptomycin, 10 mM HEPES, and 2% FCS) with 200 ml of 0.5 M EDTA added at the time of dissociation. Tissue was incubated in dissociation solution for 15 min at 37°C with rotating. Cells were then incubated on ice for 10 min and then shaken 15–20 times. At this time, microscopic examination revealed that crypts had been released and therefore the large pieces of tissue were removed. The crypts were then spun down at 500× *g* for 5 min. The supernatant was removed, and crypts were washed one time in cold PBS. Crypts were then digested to single cells by incubation with TrpLE for 5–10 min at 37°C or until microscopic examination showed single cells were present. Cells were then spun at 500× *g* for 5 min and washed one time with PBS. Any red blood cell contamination was removed by a 3 min incubation on ice with ACK lysis buffer (Thermo). Cells were then spun at 500× *g* for 5 min and washed one time with PBS. Supernatant was removed, and the cell pellet was resuspended in PBS supplemented with 0.04% BSA and passed through a 40-μm cell strainer and used directly for single-cell RNA-Seq.

For organoid samples, cultured organoids harvested after 0 (mock), 4, and 16 h post-infection were washed in cold PBS to remove excess Matrigel and incubated in TrypLE Express for 25 min at 37°C. When microscopic examination revealed that cells had reached a single-cell state, they were resuspended in DMEM/F12 and spun at 500× *g* for 5 min. Supernatant was removed, and the cell pellet was resuspended in PBS supplemented with 0.04% BSA and passed through a 40-μm cell strainer. Resulting cell suspensions were either stained with DAPI (BD Biosciences, 1:1,000) for dead cell labeling and FACS sorted (AriaFusion, BD Biosciences) or used directly for single-cell RNA-Seq.

## Single-cell RNA-Seq library preparation

Single-cell suspensions were loaded onto the 10x Chromium controller using the 10× Genomics Single-Cell 3′ Library Kit v2 (10× Genomics) and Genomics Single-Cell 3′ Library Kit v3.1 NextGem (10× Genomics) according to the manufacturer's instructions. In summary, cell and bead, emulsions were generated, followed by reverse transcription, cDNA amplification, fragmentation, and ligation with adaptors followed by sample index PCR. Resulting libraries were quality checked by Qubit and Bioanalyzer, pooled, and sequenced using NextSeq500 (Illumina; high-output mode, paired-end 26 × 75 bp). Technical validation (Appendix Fig S6) was also performed to show that sorting and fixing of the organoids can be done without affecting the general transcriptomic profile.

## Pre-processing and quality control of scRNA-Seq data

Raw sequencing data were processed using the CellRanger software (version 3.1.0). Reads were aligned to a custom reference genome created with the reference human genome (GRCh38) and human astrovirus (NC_001943.1). The resulting unique molecular identifier (UMI) count matrices were imported into R (version 3.6.2) and processed with the R package Seurat (version 3.1.3) (Stuart *et al*, 2019) Low-quality cells were removed, based on the following criteria. First, we required a high percentage of mitochondrial gene reads. For the organoid data, all the cells with mitochondrial reads > 10% were excluded. For the tissue data, all the cells with mitochondrial reads > 20% were removed. Second, we limited the acceptable numbers of detected genes. For both types of samples, cells with < 600 or > 5,000 detected genes were discarded. The remaining data were further processed using Seurat. To account for differences in sequencing depth across cells, UMI counts were normalized and scaled using regularized negative binomial regression as part of the package *sctransform* (Hafemeister & Satija, 2019). Afterward, ileum biopsies and organoids samples were integrated independently to minimize the batch and experimental variability effect. Count matrices from (Wang *et al*, 2020) were downloaded from NCBI using the accession number GSE125970 and integrated with the ileum biopsies dataset. The integration was performed using the *IntegrateData* function from Seurat. (Haghverdi *et al*, 2018; Stuart *et al*, 2019). The resulting corrected counts were used for visualization and clustering downstream analysis and non-integrated normalized counts for any quantitative comparison.

## Clustering and identification of cell type markers

We performed principal component analysis (PCA) using 3,000 highly variable genes (based on average expression and dispersion for each gene). The top 30 principal components were used to construct a shared nearest neighbor (SNN) graph and modularity-based clustering using the Louvain algorithm was performed. Finally, Uniform manifold approximation and projection (UMAP) visualization was calculated using 30 neighboring points for the local approximation of the manifold structure. Marker genes for every cell type were identified by comparing the expression of each gene in a given against the rest of the cells using the receiver operating characteristic (ROC) test. To evaluate which genes classify a cell type, the markers were selected as those with the highest classification power defined by the AUC (area under the ROC curve). These markers along with canonical markers for intestinal cells were used to annotate each of the clusters of the ileum samples (Appendix Figs S2A and S3A). For the organoids, the Seurat label transfer routine (Butler *et al*, 2018) was used to map the cell types from the ileum to the organoid's cells (Appendix Fig S3B–D). Beforehand, immune and stromal cells were filtered and only epithelial cells were used for the ileum reference. Annotation of the organoids cell types was then manually curated using the unsupervised clusters and the identified marker genes.

## Pseudotime inference

To reconstruct possible cell lineages from our ileum single-cell gene expression data, we included only the cell types from the connected epithelial lineages. The UMAP embedding was then used as input for pseudotime analysis by slingshot (Street *et al*, 2018). Stem cell was used as a start cluster and Enterocyte and Goblet as the expected end cluster. After pseudotime, time was determined and cells were ordered through the resulting principal curves. The genes that significantly change through the pseudotime were determined by fitting a generalized additive model (GAM) for each gene and testing the null hypothesis that all smoother coefficients within the lineage are equal, using the TradeSeq package (Van den Berge *et al*, 2020). Significantly changing genes were plotted in a heatmap using pheatmap (v. 1.0.10), with each column representing a cell ordered through the pseudotime of either lineage.

## Differential expression analysis

To identify the changes in expression across conditions, we performed differential expression test using MAST (Finak *et al*, 2015), which fits a hurdle model to the expression of each gene, using logistic regression for the gene detection and linear regression for the gene expression level. To reduce the size of the inference problem and avoid cell proportion bias, separate models were fit for each cell lineage and comparisons between mock, 4 h, and 16 h post-infection were performed. False discovery rate (FDR) was calculated by the Benjamini–Hochberg method (Benjamini & Hochberg, 1995), and significant genes were set as those with FDR of < 0.05. Subsequently, interferon-related genes that significantly changed across the conditions (FDR < 0.05) were used to calculate PCA, unsupervised graph-based clustering, UMAP visualization, and heatmap. Genes whose mRNAs were found to be differentially expressed were subjected to a gene set overrepresentation analysis using the EnrichR package in R (Kuleshov *et al*, 2016).

## Multiplex RNA FISH sample preparation

For multiplex FISH in 3D, organoids were infected, harvested after 16 hpi, and embedded in OCT. Section of 10 µm was cut on a cryostat (Leica) and stored at −80 until use. For multiplex FISH in 2D, organoids seeded at 60–70% confluence were infected and harvested after mock treatment or 4 and 16 hpi. HiPlex (RNAscope) was performed following the manufacturer's instruction. The RNAscope HiPlex Assay uses a novel and proprietary method of in situ hybridizations (ISH) to simultaneously visualize up to 12 different RNA targets per cell in samples mounted on slides. Briefly, sections and 2D seeded organoids were fixed in 4% paraformaldehyde, dehydrated with 50, 70, 100% ethanol, then treated with protease. All the HiPlex probes were hybridized and amplified together. Probes were designed for genes identified as cell type marker and/or corroborated by literature. The probes used for the 3D multiplex RNA FISH can be found in Appendix Table S3 and the probes used for the 2D multiplex RNA FISH in Appendix Table S4.

The detection was performed iteratively in groups of four targets. After washing, cell nuclei were counterstained with DAPI, and samples were mounted using ProLong Gold Antifade Mountant. Imaging was performed with the camera Nikon DS-Qi2 (Nikon Instruments) with the Plan Fluor 40× objective for 3D multiplex FISH and 20× objective for 2D multiplex FISH (Nikon Instruments) mounted on the Nikon Ti-E inverted microscope (Nikon Instruments) in bright-field and fluorescence (DAPI, GFP, Cy3, Cy5, and

Cy7 channels). The microscope was controlled using the Nikon NIS Elements software. After each round, fluorophores were cleaved and samples moved on to the next round of the fluorophore detection procedures. All images from all rounds of staining were then registered to each other to generate 15 plex images using HiPlex image registration software (ACD Bio). Further brightness and contrast adjustments were performed using Fiji (Schindelin *et al*, 2012).

**Multiplex RNA FISH data analysis**

The HiPlex probe fluorescent signal of 2D seeded organoids was used to quantitatively determine the cell type, the expression of innate immune transcripts, and HAstV infection levels. To obtain a resolution at a single-cell level, first nuclei segmentation and classification was done on raw DAPI images using the Pixel Classification + Object Classification workflow from ilastik 1.2.0. The resulting Object Prediction masks represented all nuclei as individual objects in a 2D plane and were saved as 16bit Tagged Image File Format.

To reduce illumination errors, a background subtraction with a rolling ball radius of 100 pixels using Fiji was performed on the raw grayscale images for APOA4, CCL2, FABP2, IFI27, IFNGR1, IRF2, MKI67, OLFM4, and STAT1 probes. To measure the single-cell fluorescent intensity for all probes, a pipeline using CellProfiler 3.1.9 was developed. Briefly, first the grayscale images corresponding to the probe fluorescent signals were uploaded on the pipeline. These images were specified as images to be measured. The corresponding Object Prediction masks previously generated by ilastik were then uploaded, converted into binary nuclei masks, and used to define the objects to be measured. Finally, with a MeasureObjectIntesity module the fluorescence intensity features, the cell number and the single-cell localization were measured for the identified objects from the binary nuclei mask.

The outcome was exported to a spreadsheet and contained the localization as well as the mean, minimum, and maximum intensity units rescaled from 0 to 1 for all probe fluorescent signal for each single cell. Nuclei wrongly illuminated, which could lead to false-positive or false-negative signal, were eliminated with a function based on the maximum and minimum fluorescent intensity within each nucleus. To determine the infection status for every cell, a threshold was calculated using the HAstV1 mean fluorescent intensity signal of mock-treated versus representative infected cells. The cell lineage was determined using the fluorescence intensity of stem cell marker OLFM4, the goblet cell marker FCGBP, the enterocyte lineage marker FABP6, and the mature enterocyte marker APOA4. For each cell type marker, a threshold was calculated based on the average cell type marker fluorescent intensity and its standard deviation. The accuracy of the threshold was controlled manually with random images. Finally, GraphPad Prism 8 was used to plot mean intensity fluorescent values against each other, or against different infected and bystander organoid cell types.

# Data and code availability

The raw sequencing and count matrices generated during this study are available at the NCBI Gene Expression Omnibus (accession no. GSE171620). https://www.ncbi.nlm.nih.gov/geo/query/acc.cgi?acc = GSE171620.

**Expanded View** for this article is available online.

## Acknowledgements

This work was supported by research grants from the Deutsche Forschungsgemeinschaft (DFG): project numbers 415089553 (Heisenberg program), 240245660 (SFB1129), 278001972 (TRR186), and 272983813 (TRR179) to SB. MS was supported by the DFG (416072091). CMZ is supported by the SFB1129 (240245660). We also acknowledge funding from the Helmholtz International Graduate School for Cancer Research to CK, Darwin Trust of Edinburgh to ST, and the ERC Consolidator Grant METACELL (773089) to TA. Open Access funding enabled and organized by Projekt DEAL.

## Author contributions

ST and MLS performed experiments, analyzed the data and contributed to manuscript writing. CM-Z analyzed the microscopy data. CK, MS, MM, MP, DO-R, and VB performed experiments. RK provided biopsy samples for the work. CS analyzed data. SB and TA conceived experiments, interpreted results and wrote the manuscript. The final version of the manuscript was approved by all authors.

## Conflict of interest

The authors declare that they have no conflict of interest.

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
