## [Review Process File · Molecular Systems Biology]

Single-cell transcriptomics reveals immune response of intestinal cell types to viral infection

Author information redacted

DOI: [10.15252/msb.20209833](https://doi.org/10.15252/msb.20209833)

Corresponding author(s): Author information redacted Author information redacted (Author information redacted) Author information redacted

Review Timeline:

Submission Date:	1st Jul 20
Editorial Decision:	6th Aug 20
Revision Received:	17th Apr 21
Editorial Decision:	18th May 21
Revision Received:	17th Jun 21
Accepted:	18th Jun 21

Editor: Jingyi Hou

Transaction Report:

Thank you for submitting your work to Molecular Systems Biology. We have now heard back from the three reviewers who agreed to evaluate your manuscript. As you will see below, the reviewers acknowledge the potential interest of the study. They raise however a series of concerns, which we would ask you to address in a major revision.

I think that the reviewers' recommendations are rather clear and there is no need to reiterate the comments listed below. Importantly, reviewers pointed out that the presented novel biological insights remain rather modest. They also mentioned that additional experiments and analyses would be required to improve the biological insights and to strengthen the manuscript.

All other issues need to be addressed as well. As you may already know, our editorial policy allows in principle a single round of major revision, so it is essential to provide responses to the reviewers' comments that are as complete as possible. Please feel free to contact me in case you would like to discuss in further detail any of the issues raised by the reviewers.

On a more editorial level, please do the following:

REFEREE REPORTS

Reviewer #1:

In this manuscript, the authors ambition to decompose the infection of the intestinal epithelial layer by the enteric virus Astrovirus (HAstV1) at the single cell level. The author aim (i) to propose a scRNA-seq Atlas of the human ileum, (ii) to form and characterize organoids derived from the intestinal cells, (iii) to characterize the infection of organoids with HAstV1 using scRNA-seq.

The paper has a strong potential because it brings a human ex vivo model to study human intestinal infections. However as presented it fails to provide robustness on the experiments because of lack of biological replicates and the overall quality of the message is severely impaired in my opinion. The paper could have a large audience including virologist of course, but also the community interested in intestinal biology and infection. The use of scRNA-seq is very interesting and the dataset presented have a huge potential but it remains here superficial in many aspects (see major comments). Finally, the study of infection using scRNA-seq suffers from major holes that needs serious improvement.

Major comments

1. The Introduction should be completed to add the essential knowledge on the astrovirus biology. The Virus tropism has been investigated using a mouse model including using single cell RNA-seq (Cortez et al 2020 Nat Comm). As this study challenge the use of mice models, the limitation of the latter should be briefly discussed.
2. Figure 1 and S1: the immunofluorescence panels are very small and scales are missing. The

quality of the figure can be largely improved.

3. In Figure S1 (infection of cell lines Caco-2 and T84) we see a massive increase of infected cell percentage to reach 100% after 16 hours post-infection (hpi, Figure S1B). How this increase is explained? The percentage of infected cells in Figure 1 (intestinal organoids) should also be clarified and the kinetics of the infection (percentage of infected cells over time).

4. Figure 2 claims to reach an 'Atlas' of the human ileum biopsies. The number of biopsies should be clearly stated in the main text (page 5). Figure S2A should also be clarified so the reader can understand what is the difference between 'Ileum1_a and 1_b'.

5. Figure 2: the authors discuss very poorly in the text the genes that allowed them to assign the different cell populations. Such information has been partially moved in the Method section but should be discussed in the text in-depth. The text needs to be improved in that respect.

6. Figure 2A assign populations as 'progenitors' or 'immature' and is back up by Figure 2C&D that uses a pseudotime analysis as a definitive to see 'gradients' and 'waves' of gene expression. Since the authors aim that these data are viewed as 'a comprehensive' and 'a cell-annotated reference single cell atlas of human ileum', the position of the different cell types should be provided precisely using human material (using in situ hybridization for example) and get as claimed 'a comprehensive atlas'. Data analysis should be strengthened using RaceID and diffusion maps (as example, the authors can use Aizarani et al 2019 Nature). Ideally if the author wants to claim that populations are 'progenitors', they should be culturing the cells and specifically demonstrate their potential to generate the downstream cell types.

7. Figure 3 and Figure S4 aim to demonstrate that ileum derived organoids are matching the initial tissue. The match between tissue vs organoids needs a serious improvement. First a suppl Figure should recapitulate the expression of main cell type markers (text book knowledge) across the organoids. Secondly, what is called 'immature enterocyte' and 'Enterocyte progenitor' in Figure 2 do not match. For example: gene markers associated to 'enterocyte progenitor' like ADIRF, FABP1, GSTA1 do not mark this population. It belongs to the authors to clearly demonstrate what is matching or not, and henceforth clearly outline their demonstration.

8. Figure 4 shows data from infected organoids with HAstV1. The overall percentage of infected cells is extremely low, app. 1%. Consequently, the authors recover a very limited number of infected cells - I have counted less than 100 on Figure 4A. In addition, this experiment seems to come from a unique experiment (n=1) as far we can judge from the data. Figure 4A and B again fail to present all the claim data: while the text promises us 4 and 16 hpi, here the figure is undetermined. The data presented are very problematic to draw solid conclusions. If the virus could be found in diverse cells, not sure that the virus replicates in all cell types. These data seem to clearly conflict with the data from Figure 1 that gave the impression that a much higher infection rate was reached. Figure 1D claimed a 100 fold increase of HAstV1 copy number between 4 and 16 hpi. The single cell data should recapitulate this result. Also, the data presented conflict with the recently published from Cortez et al (2020 Nat Comm; <https://doi.org/10.1038/s41467-020-15999-y>) that claims that the virus is present mainly in goblet cells. This should be discussed.

9. A strong claim of the study is that ISG are ISG (Interferon stimulated genes) are differently expressed between the different cell types. (i) It is not clear if the authors look at infected cells only or all cells including bystanders. (ii) in the absence of clear biological replicates, we cannot judge of

the robustness of the claims. (iii) The authors claims differential expression of ISG at basal level. From our own experience, we noticed that ISG expression can vary dramatically from one experiment to the other. It emphasizes again the need to have robust biological replicates.

Minor comments

- Figure S2A: typo 'Ileum' - only one 'l'
- Figure S2E: the x axis can be log transformed
- Figure S2A and D-F: the authors should have a consistent color code for their samples.
- Legend Fig 4E. typo with two '.' at end of the sentence.
- The method to expand the virus should be expanded.

Reviewer #2:

The manuscript, "Single-cell transcriptomics reveals immune response of intestinal cell types to viral infection" uses single-cell RNA-Seq to investigate astrovirus infection of human ileum derived biopsies and intestinal organoids. Here astrovirus was found to infect all major cell types, each of which undergoes a unique 'anti-viral' transcription program upon infection. This is marked by unique ISG expression patterns. Multiplex RNA imaging after differentiation of the various identified cell types within organoids allowed the authors to extend their findings spatially. These results indicated that cell-type specific ISG expression was largely lineage specific suggesting cell-lineage specific anti-viral responses.

This is an interesting well-performed study. The manuscript is clearly written and does justice to recently published work that shares a large degree of overlap. Beyond extending recent scRNA-Seq datasets (Wang et al. 2020) from human ileum biopsies, by combining multiplex RNA imaging the manuscript allows for direct spatial correlation of cell-type specific ISG expression, and virus infection.

My only concern is the limited advance in understanding ISG mediated control of astroviruses, even in an organoid setting. Deconvolution of cell-types (individual cell lines) and the impact of variable ISG-mediated control of astrovirus infection would be warranted here to support the authors claims. Do different cell types control the virus differently (outcome of interferon response)? Do cell types display variant infection outcomes due to loss of different ISGs? Such data would provide solid validation for the utility of this combined single-cell RNA-Seq/ Multiplex RNA imaging approach

Reviewer #3:

In this manuscript, the authors do two things; undertake single cell transcriptomics to identify cell populations within the ileum and define astrovirus induction of interferon in different ileal organoid cell populations. There are many strengths in the manuscript including the importance of the topic, use of multiplex in situ hybridization to visualize multi-cellular organization of organoids, and the potential identification of unique enterocyte populations within the ileum. However, the concerns about validation of the newly-defined enterocyte populations, methodology, and the novelty of the astrovirus data do dampen enthusiasm for this potentially interesting study.

Major concerns:

1. While interesting, the astrovirus data primarily confirms what is in the literature. The authors

should consider highlighting the novelty of their findings for the reader.

2. In the organoids, was the upregulated ISGs during infection in astrovirus infected or bystander cells or both? Please discuss.
3. COMPUTATIONALLY-defining 14 cell clusters in a dataset does not make the results more robust since validation work was not performed to examine the biological relevance/true existence. For example, the authors should provide further information on what an "enterocyte 2" cells is and it's function.
4. If the authors subtract out immune and stromal cells from their dataset, do they have a comparable 6000 cells like the Wang et al 2020 dataset? How many cells total were identified in each of the 14 cell populations?
5. Given what look like low numbers for some of the populations, can the authors be certain that these are truly distinct cell populations? The study may still be underpowered to determine this. To strengthen the conclusions and provide further validation of the 14 cell types denoted in their dataset, the authors should aggregate the Wang et al dataset to see if they can better validate their findings.
6. The correlation scores between the enteroid and ileal cell types are not very high. How many ileal biopsies and enteroids were used for the studies and were these from distinct donors? Based on the correlation matrices, one could argue that certain cell populations were classified with their best match in the ileal dataset (eg. Enterocyte progenitors in the enteroids looking more like TA cells from the biopsies, secretory progenitors looking more like stem cells or Cycling TA).
7. The Best4+ enterocyte population found in enteroids appears to be highly expressing lysozyme (Fig 3b), which is generally a Paneth cell-specific marker. Are the authors confident in their classifications?
8. While much appreciated, having the ileal biopsy dataset to "validate" organoid datasets may not be as critical as the authors describe.
9. Given the low level of KI67 expression across all cells (Fig. 3b), it is unclear whether the RNA-specific probe data showing the preferential infection of KI67+ cells is well-supported. Is it possible that the virus is inducing this cell proliferative marker?
10. Kolawole et al saw that enterocytes, goblet cells and progenitor cells were the main cell types infected, which is inconsistent with these results. Please discuss.
11. The single-cell data representation of HAstV1 infection is described with the binary plot (Fig. 4a), which does not show the level of expression of the virus. This is critical since free-floating viral RNA could contaminate gel beads and produce an artifact that would look like infection. Thus, it is unclear where the virus is truly replicating. The authors must address this point as it is critical to data interpretation.
12. The level of infection is also quite low overall (Fig. 4a & b), which is a major limitation since the authors are unable to compare infected versus uninfected cells within each cell population to truly look at the IFN responses between infected and bystander cells. Again, this must be reconciled to support the authors conclusions.
13. IFN responses were compared at 4 and 16 hpi relative to mock infection, which was not taken at the same time points but was "0" hpi time point, when presumably the enteroids are reforming. Can the authors be certain that "each individual cell lineage" mounting distinct transcriptional responses are not derived from these experimental changes in the enteroids?
14. These studies could be greatly strengthened if they could show that IFN administration to infected enteroids limits infection of certain cell types? As the authors preface, stem cells are not IFN responsive and yet they report infection of stem cells. Thus, one could hypothesize that these cell types would still be infected if the virus truly shows a stem cell tropism.

Minor suggestions

1. The title should be modified to better describe the studies. It is really examining IFN-associated

changes in astrovirus treated ileal-derived enteroids.

2. Using 2 techniques typically does not represent a pipeline. Please clearly define how this is a new pipeline.

3. The authors use the term "organoid" throughout but these HIEs were derived from stem cells and thus, only contain epithelial cells. Organoid is used to define cultures derived from iPSCs that contain mesenchymal cells. Please discuss.

4. Results-astrovirus infections do not lead to death in children and immunocompromised-that's really specific for the astro-associated CNS infections. Please clarify.

Dear Jingyi Hou and reviewers,

We would like to thank you for your reviews of our manuscript. We are presenting you our revision that addresses all raised concerns. First, we have addressed the concern about replicates by clarifying that our data was not a single replicate but we have used between n=3 to n=7 biological replicates depending on the samples (with the number of biological replicates n=7 for the mock condition, n=3 for the 4 hpi condition, and n=5 for the 16 hpi condition). We have also expanded our panel of RNAscope probes to better identify which cells are infected or bystander cells and to evaluate/control the response to astrovirus infection mounted by these cells. Finally, we repeated our scRNAseq experiment of astrovirus-infected organoids using the just-released version 3 of the 10X Genomics scRNAseq kit. Using the new version of the kit, we were able to achieve an even better detection of the virus transcripts which correlated with the amount of virus replication that we observed by both RNAscope and immunofluorescence stainings. This further supports our findings that human astrovirus 1 (HAstV1) is capable of infecting and productively replicating in all cell types in the human intestinal tract. Additionally, we have used RNAscope to evaluate how infected and bystander cells upregulate interferon-stimulated genes (ISGs). Using this orthogonal approach to scRNAseq, we could determine that both infected and bystander cells produce similar ISGs however the magnitude of their induction is different and infected cells induced a higher amount of ISGs compared to bystander cells.

Please find below our point by point responses to the reviewers comments. The reviewer comments are in **black italic** and our responses are in **blue**.

Reviewer #1:

In this manuscript, the authors ambition to decompose the infection of the intestinal epithelial layer by the enteric virus Astrovirus (HAstV1) at the single cell level. The author aim (i) to propose a scRNA-seq Atlas of the human ileum, (ii) to form and characterize organoids derived from the intestinal cells, (iii) to characterize the infection of organoids with HAstV1 using scRNA-seq.

*The paper has a strong potential because it brings a human ex vivo model to study human intestinal infections. However as presented it fails to provide **robustness on the experiments because of lack of biological replicates** and the overall quality of the message is severely impaired in my opinion. The paper could have a large audience including virologist of course, but also the community interested in intestinal biology and infection. The use of scRNA-seq is very interesting and the dataset presented have a huge potential but it remains here superficial in many aspects (see major comments). Finally, the study of infection using scRNA-seq suffers from major holes that needs serious improvement.*

We would like to thank the reviewer for their comments. We are sorry that it was not clear that the results presented were done over at least n=3 biological replicates (with the

number of biological replicates n=7 for the mock condition, n=3 for the 4 hpi condition, and n=5 for the 16 hpi condition) and therefore provide robust data. We have also added additional replicates during the revision process and have clarified this in the text so that it is clear that our data comes from at least n=3 biological replicates. Please see below our point-by-point answers and our explanations of the modifications that we have provided during the revision process. We hope that this new version clarifies your concerns and that it is now more accessible for a broad audience.

Major comments

1. *The Introduction should be completed to add the essential knowledge on the astrovirus biology. The Virus tropism has been investigated using a mouse model including using single cell RNA-seq (Cortez et al 2020 Nat Comm). As this study challenge the use of mice models, the limitation of the latter should be briefly discussed.*

The introduction has been updated to include more information and references about astrovirus and its worldwide prevalence. Human and murine viruses are very different and often behave differently in their host. While Cortez et al clearly show that murine astrovirus replicates in goblet cells, previous work using human biopsies has shown that human astrovirus replicates in enterocytes again showing that murine and human viruses can have distinct cell tropisms. A discussion on this has been added to the text, with references included. We made sure that our text does not read like we are challenging the findings by Cortez et al., but simply highlighting the potential differences between murine and human astrovirus. We also made a statement that our work uses human organoids which, although they are the best model system available, they still have their limitations and this could explain some of the observed differences between our study (organoids) and the Cortez study (animal mice).

2. *Figure 1 and S1: the immunofluorescence panels are very small and scales are missing. The quality of the figure can be largely improved.*

The immunofluorescence images have been enlarged and scale bars have been added to the images. Figure S1 is now called Extended view EV1.

3. *In Figure S1 (infection of cell lines Caco-2 and T84) we see a **massive increase of infected cell percentage to reach 100% after 16 hours post-infection (hpi, Figure S1B)**. How this increase is explained? The percentage of infected cells in Figure 1 (intestinal organoids) should also be clarified and the kinetics of the infection (percentage of infected cells over time).*

The astrovirus lifecycle in these cell lines is around 10-16 hours. The increase in infection at 16h can be due to either the release of virus from the first round of infection and the start of second rounds of infection or to the fact that the primary infection is not synchronised and that some cells can take much longer to be infected. Additionally, we have added the replication kinetics and the percentage of infected cells in the intestinal organoids to Figure 1 (Fig.1D).

4. *Figure 2 claims to reach an 'Atlas' of the human ileum biopsies. The number of biopsies should be clearly stated in the main text (page 5). Figure S2A should also be clarified so the reader can understand what is the difference between 'Ileum1_a and 1_b'.*

We thank the reviewer for their comments and have updated the manuscript to show that we used two human biopsies with two technical replicates each as reference samples. At the time when this work was originally performed there was not an “Atlas” available to assign cell types for the human Ileum. There are now some preprints available where “proper” atlases have been generated, however their data is still deposited and publically available. Therefore, we used the biopsies from our gastroenterologist to make our own reference. As we only had two biopsies, we have now used the Wang reference set as well and have updated the manuscript to note that these are “reference data sets” and not “Atlases”.

5. *Figure 2: the authors discuss very poorly in the text the genes that allowed them to assign the different cell populations. Such information has been partially moved in the Method section but should be discussed in the text in-depth. The text needs to be improved in that respect.*

We added Appendix Figure S2, where we provide details on the markers used for the annotation of each cell type and the associations between the unsupervised clusters and the final annotation. We have also updated the text to make this more clear.

6. *Figure 2A assign populations as 'progenitors' or 'immature' and is back up by Figure 2C&D that uses a pseudotime analysis as a definitive to see 'gradients' and 'waves' of gene expression. Since the authors aim that these data are viewed as 'a comprehensive' and 'a cell-annotated reference single cell atlas of human ileum', the position of the different cell types should be provided precisely using human material (using in situ hybridization for example) and get as claimed 'a comprehensive atlas'. Data analysis should be strengthen using RaceID and diffusion maps (as example, the authors can use Aizarani et al 2019 Nature). Ideally if the author wants to claim that populations are 'progenitors', they should be culturing the cells and specifically demonstrate their potential to generate the downstream cell types.*

In order to improve our population assignment and to create a more comprehensive reference, we used the ileum dataset from Wang et al 2020 in addition to our biopsy samples. We now do claim to create an Atlas which we agree was a bit of an overstatement in the original submission. We have also included in Figure 2 the change of expression along the pseudo time of two lineage markers for both major lineages. Concerning the proposal of using RaceID, actually the slingshot tool we used is very similar to the fateBias approach used in RaceID. So, we believe that we have this analysis covered with slingshot and using RaceID would not bring any advantages. We also believe that diffusion maps are obsolete when compared to UMAPs as the diffusion maps tend to lose the population structure during differentiation processes. Following your request, we, however, evaluated the use of diffusion maps, and found that it does not provide any advantages compared to the UMAP used by us. Please see the plot below (not included in the manuscript) illustrating the limitation of the diffusion map for our data set.

7. Figure 3 and Figure S4 aim to demonstrate that ileum derived organoids are matching the initial tissue. The match between tissue vs organoids needs a serious improvement. First a suppl Figure should recapitulate the expression of main cell type markers (text book knowledge) across the organoids. Secondly, what is called 'immature enterocyte' and 'Enterocyte progenitor' in Figure 2 do not match. For example: gene markers associated to 'enterocyte progenitor' like *ADIRF*, *FABP1*, *GSTA1* do not mark this population. It belongs to the authors to clearly demonstrate what is matching or not, and henceforth clearly outline their demonstration.

As suggested by the reviewer, we have added Appendix Figure S3 that recapitulates the expression of main cell type markers across the organoids. We have improved the annotation so this is consistent between the tissues and the organoids. Now the various cell types identified in the ileum and their associated marker genes are matching with ones identified in our organoids (Appendix Fig. S2 vs S3).

8. Figure 4 shows data from infected organoids with HAdV1. The overall percentage of infected cells is extremely low, app. 1%. Consequently, the authors recover a very limited number of infected cells - I have counted less than 100 on Figure 4A. In addition, this experiment seems to come from a unique experiment ($n=1$) as far we can judge from the data. Figure 4A and B again fail to present all the claim data: while the text promises us 4 and 16 hpi, here the figure is undetermined.

The data presented are very problematic to draw solid conclusions. If the virus could be found in diverse cells, not sure that the virus replicates in all cell types. These data seem to clearly conflict with the data from Figure 1 that gave the impression that a much higher infection rate was reached. Figure 1D claimed a 100 fold increase of HAdV1 copy number between 4 and 16 hpi. The single cell data should recapitulate this result.

Also, the data presented conflict with the recently published from Cortez et al (2020 Nat Comm; <https://doi.org/10.1038/s41467-020-15999-y>) that claims that the virus is present mainly in goblet cells. This should be discussed.

We would like to emphasize (see our earlier response) that in fact our data are not from a single replicate but from 3 to 7 biological replicates depending on the samples. We, however, understand the concern of the reviewer about the low numbers of the infected cells. To provide

a robust response to the reviewer, we have now repeated our scRNAseq experiment of astrovirus infected human organoids. In the original manuscript, we used the version 2 of the kit that indeed provided only a limited detection of virus transcripts. In this revised version of the manuscript, we have performed new experiments using the recently released 10x Genomics kit v3.1 NextGem, where the capture rate is higher and more efficient. This provided a higher dynamic range and allowed us to detect the infected cells in a proportion consistent with the virus detection by immunofluorescence in Figure 1 and RNAscope in Figure 3. We have updated Figure 4 with the new data (10x Genomics kit v3.1) for 4 and 16 hpi. In short we believe that our low detection of virus infected cells in the original version which did not match with the immunofluorescence staining was due to the version 2 of the 10X Genomics kit. IN this revised version, we now have a good match between the percentage of infected cells as identified using immunofluorescence and RNAscope and the percentage determined from the scRNAseq. (40% by IF in Fig1D, ~40% by scRNAseq in Fig 4B and ~40% by RNAscope, statement in text).

(Cortez et al 2020 Nat Comm) indeed shows that murine astrovirus replicates in goblet cells and that there are low levels of virus in all other cell types. Our work shows that we have active virus replication in all cell types and we see a large increase in the number of positive cells in all cell lineages between 4 and 16 hours (new Figure 4B and Appendix Fig S5) yet with clear differences between cell types, that further confirms the effect of virus replication as our data cannot be explained by background contamination. The number of infected cells in Cortez et al was also rather low. Since we have undoubtedly established that 10x Genomics kit v2 has insufficient sensitivity (or other limitation resulting in a low rate of sequencing of viral transcript) whereas 10x Genomics kit v3 provides a substantial increase in sensitivity with the results consistent with two other orthogonal technologies (immunofluorescence, RNAscope FISH). There is a possibility that Cortez et al also faced the same limitations as they also used 10x Genomics kit v2. However, as we now have highlighted in our discussion, the Cortez study uses murine Astrovirus and mouse model and this could be intrinsically different to the human Astrovirus. Similarly, we also discussed that human organoids have their limitations and this could lead to a greater infectivity rate compared to real in-vivo models.

9. A strong claim of the study is that ISG are ISG (Interferon stimulated genes) are differently expressed between the different cell types. (i) It is not clear if the authors look at infected cells only or all cells including bystanders. (ii) in the absence of clear biological replicates, we cannot judge of the robustness of the claims. (iii) The authors claims differential expression of ISG at basal level. From our own experience, we noticed that ISG expression can vary dramatically from one experiment to the other. It emphasizes again the need to have robust biological replicates.

We thank the reviewer for raising these key points. (i) The ISG expression was determined from all cells (infected and bystanders) in the 4hpi and 16hpi conditions compared to the mock condition. (ii) As clarified in our earlier responses, our study did not result from a single experiment but was performed from 3 to 7 biological replicates (Appendix Figure S1) and the results have been consistent between the replicates. Importantly, we have now performed an additional scRNAseq experiment using 10x Genomics kit v3 and our results again show that

individual cell types generate a different immune response (the new Fig. 4 includes these new datasets). As an orthogonal approach, we have exploited RNAscope and can show that different cell types express a different “pattern” of ISGs (new Fig. 5 and Appendix Figures S6-7) (iii) We agree with the reviewer and also noticed that the magnitude of ISG expression (induced or basal) can vary between replicates however, this does not affect which ISGs are more predominant in a stem cell vs an enterocyte comparison as our differential gene expression analysis using MAST included all replicates and took into account the samples variability.

Minor comments

- *Figure S2A: typo 'Illeum' - only one 'l'*

This figure has been updated.

- *Figure S2E: the x axis can be log transformed*

This axis has been updated.

- *Figure S2A and D-F: the authors should have a consistent color code for their samples.*

The color code has been updated.

- *Legend Fig 4E. typo with two '..' at end of the sentence.*

This has been updated in the text.

- *The method to expand the virus should be expanded.*

The methods section has been expanded to show the passage of the virus and the method used to calculate MOI.

Reviewer #2:

The manuscript, "Single-cell transcriptomics reveals immune response of intestinal cell types to viral infection" uses single-cell RNA-Seq to investigate astrovirus infection of human ileum derived biopsies and intestinal organoids. Here astrovirus was found to infect all major cell types, each of which undergoes a unique 'anti-viral' transcription program upon infection. This is marked by unique ISG expression patterns. Multiplex RNA imaging after differentiation of the various identified cell types within organoids allowed the authors to extend their findings spatially. These results indicated that cell-type specific ISG expression was largely lineage specific suggesting cell-lineage specific anti-viral responses.

This is an interesting well-performed study. The manuscript is clearly written and does justice to recently published work that shares a large degree of overlap. Beyond extending recent scRNA-Seq datasets (Wang et al. 2020) from human ileum biopsies, by combining multiplex RNA imaging the manuscript allows for direct spatial correlation of cell-type specific ISG expression, and virus infection.

*My only concern is the **limited advance in understanding ISG mediated control of astroviruses**, even in an organoid setting. Deconvolution of cell-types (individual cell lines) and the impact of variable ISG-mediated control of astrovirus infection would be warranted here to support the authors claims. Do different cell types control the virus differently (outcome of interferon response)? Do cell types display variant infection outcomes due to loss of different ISGs? Such data would provide solid validation for the utility of this combined single-cell RNA-Seq/ Multiplex RNA imaging approach*

We thank the reviewer for their encouraging comments and their interest and support of our work. We have now updated the manuscript by adding novel data from repeating the scRNAseq by using the recently released 10x Genomics kit (version 3). Using this kit we were better able to identify infected cells and the percentage of infected cells as identified using our scRNAseq data is now consistent with two other methods (immunofluorescence, RNAscope). In the original manuscript, we used the version 2 of the kit that indeed provided only a limited detection of virus transcripts. In this revised version of the manuscript, we have performed new experiments using the recently released 10x Genomics kit v3.1 NextGem, where the capture rate is higher and more efficient. This provided a higher dynamic range and allowed us to detect the infected cells in a proportion consistent with the virus detection by immunofluorescence in Figure 1 and RNAscope in Figure 3. We have updated Figure 4 with the new data (10x Genomics kit v3.1) for 4 and 16 hpi. In short we believe that our low detection of virus infected cells in the original version which did not match with the immunofluorescence staining was due to the version 2 of the 10X Genomics kit. In this revised version, we now have a good match between the percentage of infected cells as identified using immunofluorescence and RNAscope and the percentage determined from the scRNAseq. (40% by IF in Fig1D, ~40% by scRNAseq in Fig 4B and ~40% by RNAscope, statement in text). With these new results, we could confidently confirm that all cell types support astrovirus replication. Additionally, we have investigated infected vs bystander responses using RNAscope. This allowed us to confirm that both infected and bystander cells upregulate similar ISGs upon astrovirus infection (Figure 5 and Appendix Fig S6-7).

Reviewer #3:

In this manuscript, the authors do two things; undertake single cell transcriptomics to identify cell populations within the ileum and define astrovirus induction of interferon in different ileal organoid cell populations. There are many strengths in the manuscript including the importance of the topic, use of multiplex in situ hybridization to visualize multi-cellular organization of organoids, and the potential identification of unique enterocyte populations within the ileum. However, the concerns about validation of the newly-defined enterocyte populations, methodology, and the novelty of the astrovirus data do dampen enthusiasm for this potentially interesting study.

Major concerns:

1. While interesting, the astrovirus data primarily confirms what is in the literature. The authors should consider highlighting the novelty of their findings for the reader.

We thank the reviewer for their comment as it is important that we convey a clear message to the reader. We would like to emphasize and clarify the key novelty of our findings: We have determined that different cell lineages upregulate a distinct set of ISG upon HAstV1 infection. Importantly, we could also show that the steady state signature of ISG expression is different between the different cell lineages in the intestinal epithelium. Previous work has used bulk readouts that show overall inductions of ISGs but did not assign them to cell lineages or infected vs. bystander cells. Using scRNA-Seq (10x Genomics) and multiplex RNA FISH (RNAscope) we were able to show that cell lineages have distinct basal levels of ISGs and these are induced uniquely upon enteric virus infection. We have added several new figures further highlighting these exciting discoveries and further discussed them in the text (Fig. 5 and Appendix Figures S6-7).

2. In the organoids, was the upregulated ISGs during infection in astrovirus infected or bystander cells or both? Please discuss.

The ISG expression was determined from all cells (infected and bystanders) in the 4hpi and 16hpi conditions compared to the mock condition. We have now included new RNAscope data where we compare ISG induction in both infected and bystander cells (Fig. 5). We can see that both infected cells and bystander cells upregulate similar ISGs, however infected cells often upregulate them to a higher extent.

3. COMPUTATIONALLY-defining 14 cell clusters in a dataset does not make the results more robust since validation work was not performed to examine the biological relevance/true existence. For example, the authors should provide further information on what an "enterocyte 2" cells is and it's function.

We agree with the reviewers that having more cell clusters does not equal robustness. We have updated the text and toned down this claim. Additionally, we re-evaluated the enterocyte 2 classification using both our dataset and the one from Wang et al and annotated it as Colonocytes, as this population expresses specific markers of colon enterocytes such as CA4 and CLCA4. This is now updated in Figure 2.

4. If the authors subtract out immune and stromal cells from their dataset, do they have a comparable 6000 cells like the Wang et al 2020 dataset? How many cells total were identified in each of the 14 cell populations?

We indeed do have a comparable amount of cells even without the immune and stromal cells. Please see below the amount of cells per each cell type in comparison with the Wang dataset. We have adapted the text to reflect this and have modified the text to not directly compare our data to the one from the Wang study. We rather now use the Wang dataset, together with our dataset to perform a better cell annotation.

5. Given what look like low numbers for some of the populations, can the authors be certain that these are truly distinct cell populations? The study may still be underpowered to determine this. To strengthen the conclusions and provide further validation of the 14 cell types denoted in their dataset, the authors should aggregate the Wang et al dataset to see if they can better validate their findings.

Thank you for your suggestion, we have added the two ileal datasets from Wang et al into our analysis. This indeed has improved our resolution and certainty in the cell type classification. Figure 2 was updated accordingly. Moreover, we have added a new figure (Appendix Figure S2) with details on the cell-type annotation .

6. The correlation scores between the **enteroid and ileal cell types are not very high**. How many ileal biopsies and enteroids were used for the studies and were these from distinct donors? Based on the correlation matrices, one could argue that certain cell populations were classified with their best match in the ileal dataset (eg. Enterocyte progenitors in the enteroids looking more like TA cells from the biopsies, secretory progenitors looking more like stem cells or Cycling TA).

We agree with the reviewer that organoids do not fully recapitulate the tissue and are limited to the differentiation media that they are exposed to. Unlike in the body, organoids are kept in a high stem cell state which allows them to constantly proliferate. The differentiation media that we use induces differentiation but the ratio of the absorptive vs secretory lineages is different from in vivo. Additionally, the organoids and tissue come from different donors and as such will display differences due to donor variability. As the implications of this figure are beyond the scope of the manuscript and raises more concerns we have decided to remove this figure from the manuscript.

7. The Best4+ enterocyte population found in enteroids appears to be highly expressing lysozyme (Fig 3b), which is generally a Paneth cell-specific marker. Are the authors confident in their classifications?

Yes, Best4+ enterocytes also express lysozyme but do not express other Paneth cell marker genes such as REG3A. Additionally, as you can see from our RNAscope data (Appendix Figure S4) lysozyme is expressed in most cell types and therefore it is not surprising to see it detected by scRNAseq also in the Best4+ population.

8. While much appreciated, having the ileal biopsy dataset to "validate" organoid datasets may not be as critical as the authors describe.

In order to identify our cell types, we need a reference data set to do this. At the time of this work (and currently) there was no scRNAseq atlas for the human ileum available. We therefore had to make our own reference data set using biopsies obtained from our gastroenterology department. We agree that the small number of samples used does not constitute an atlas. In this revision, to increase the confidence in the assignment of the cell types, we have integrated a published data set from (Wang et al, 2020) together with our data from the biopsies. We have also updated the text to call this a "reference data set" and not an atlas due to the small sample size.

9. Given the low level of KI67 expression across all cells (Fig. 3b), it is unclear whether the RNA-specific probe data showing the preferential infection of KI67+ cells is well-supported. Is it possible that the virus is inducing this cell proliferative marker?

After doing a high-throughput analysis using multiplex RNA FISH on 2D seeded organoids we could better evaluate the correlation between the MKI67 expression levels and the HAstV1 infection (Fig. EV2). We see that in mock-treated organoids the expression levels are very low, and they increase over time after infection. 16 hpi they reach the highest values and show a strong positive correlation with the HAstV1 signal. Moreover, the expression of MKI67 in bystanders is visibly lower than in infected cells. This new data strongly suggests that HAstV1 infection induces transcription of MKI67. This data has now been included and can be found in Figure 5.

10. Kolawole et al saw that enterocytes, goblet cells and progenitor cells were the main cell types infected, which is inconsistent with these results. Please discuss.

In the Kolawole et al paper, the authors study the cell tropism of the VA1 strain of human astrovirus which is unique as it has a neurotropism and does not require trypsin activation for its infection. Our study uses human astrovirus 1 which requires trypsin and its infection is limited to the gastrointestinal tract. As these are two different strains, it is not surprising that we see differences between the results. Nevertheless, the conclusion of both stories is very similar: Kolawole et al. and our work see a broad tropism from human astrovirus with clear infection of multiple cell types.

11. The single-cell data representation of HAstV1 infection is described with the binary plot (Fig. 4a), which does not show the level of expression of the virus. This is critical since free-floating viral RNA could contaminate gel beads and produce an artifact that would look like infection.

Thus, it is unclear where the virus is truly replicating. The authors must address this point as it is critical to data interpretation.

We have updated this Fig. 4A to show the level of expression of the virus. Using the heatmap we can now clearly see that infection distributed throughout all cell types. Additionally we added Appendix Figure S5 (see below) which shows for each cell type that a dramatic increase in viral expression from 4hpi to 16hpi thus indicating active virus replication.

12. The level of infection is also quite low overall (Fig. 4a &b), which is a major limitation since the authors are unable to compare infected versus uninfected cells within each cell population to truly look at the IFN responses between infected and bystander cells. Again, this must be reconciled to support the authors conclusions.

We thank the reviewers for their comments. In the original version of the manuscript we used the then-available 10x Genomics version 2 kit that, although allowed us to detect virus transcripts in a number of cells, delivered only limited sensitivity compared with our immunofluorescence and RNAscope data. For this revision, we have repeated the scRNAseq analyses with the version 3 kit which has greatly improved the detection of virus transcripts due to its higher dynamic range. Importantly, using the version 3 kit not only allowed us to detect more infected cells by scRNAseq. Now the numbers of virus-infected cells as detected by scRNAseq are in agreement with RNAscope and immunofluorescence results. This new data has allowed us to update our results to show the difference between bystander and infected cells.

13. IFN responses were compared at 4 and 16 hpi relative to mock infection, which was not taken at the same time points but was "0" hpi time point, when presumably the enteroids are

reforming. Can the authors be certain that "each individual cell lineage" mounting distinct transcriptional responses are not derived from these experimental changes in the enteroids?

All samples were harvested at the same time. All cells were prepared and differentiated in the same manner. For the infection, astrovirus was added to the cells 16h prior to harvesting. For the 4h infection, virus was added to the cells 4h prior to harvesting (12h after the 16h infection). The mock samples were harvested at the same time as the 4 and 16 hours. At this point all samples were processed together and loaded on the same 10X Genomics chip. The experiment was performed this way, as the batch effect that occurs between samples is affected by the processing time and the chips. Using a simultaneous harvesting avoided a potential batch effect and also allowed us to compare mock cells which were of the same age and not 16h younger to make the results more robust.

14. These studies could be greatly strengthened if they could show that IFN administration to infected enteroids limits infection of certain cells types? As the authors preface, stem cells are not IFN responsive and yet they report infection of stem cells. Thus, one could hypothesize that these cell types would still be infected if the virus truly shows a stem cell tropism.

We agree with the reviewer that this would be a very exciting question to examine. However we believe that this would go out of the scope of this manuscript. Moreover, taking into account the breadth of the raised question, we believe that performing the requested experiments with all necessary controls and validations as well as data analysis can result in a full separate manuscript.

Minor suggestions

1. The title should be modified to better describe the studies. It is really examining IFN-associated changes in astrovirus treated ileal-derived enteroids.

We would prefer to keep the title shorter and more accessible taking into account the broad audience of Molecular Systems Biology.

2. Using 2 techniques typically does not represent a pipeline. Please clearly define how this is a new pipeline.

We agree and the text has been updated to remove the term pipeline and instead refer to this as a framework to study viral infection.

3. The authors use the term "organoid" throughout but these HIEs were derived from stem cells and thus, only contain epithelial cells. Organoid is used to define cultures derived from iPSCs that contain mesenchymal cells. Please discuss.

The original paper from Clevers and Sato which isolated murine and human stem cells to generate mini-guts refers to them as organoids. This nomenclature has been kept and is still used by all papers from the Clevers and Sato labs. It seems that many labs have adopted this idea that organoids are from iPSCs and enteroids are from patients; however this nomenclature is not used by all labs world-wide. Thus, we use the nomenclature "organoid" following the original publications from the Clevers lab.

4. Results-astrovirus infections do not lead to death in children and immunocompromised-that's really specific for the astro-associated CNS infections. Please clarify.

Thank you for catching this. It is an important note to clarify. The text has been updated and the introduction has been extended to include more details about astrovirus infections and pathogenesis.

Thank you for sending us your revised manuscript. We have now heard back from the three reviewers who were asked to evaluate your study. As you will see, the reviewers are satisfied with the modifications made and think that the study is now suitable for publication.

Before we can formally accept your manuscript, we would ask you to address the following editorial-level issues.

REFEREE REPORTS

Reviewer #1:

The authors have provided a revised version of their manuscript aiming to dissect the immune intestinal response to the human astrovirus. The revision provides clarification on the number of replicates used along the manuscript and nicely enhance the publication with a new experiment aiming to better identify the infected cells. By removing the 'atlas' claim, the paper is better positioned. Overall, the revision provided a massive enhancement of the quality of the study and figures. The authors have answered all my major concerns. This publication indeed will provide an

important footprint for the field working on intestinal organoids and infection.

Minor comments

The gene names should be written in italic+capital in the figures and proteins only in capital letters.

Reviewer #2:

The authors have done a nice job of addressing my concerns, the use of the newer genomics kit with higher viral infection resolution, as well as the additional data on bystander vs infected ISG responses has served support and strenghten the original findings. I have no further suggestions for improvement.

Reviewer #3:

The authors were responsive to reviewer's queries. No further concerns.

The authors have made all requested editorial changes.

Thank you again for sending us your revised manuscript. We are now satisfied with the modifications made and I am pleased to inform you that your paper has been accepted for publication.

Corresponding Author Name: Theodore Alexandrov, Steeve Boulant

Journal Submitted to: MSB

Manuscript Number: MSB-20-9833